


# 1   The enigmatic curvature of Central Iberia and its puzzling kinematics

Daniel Pastor-Galán[1,2,3], Gabriel Gutiérrez-Alonso[4,5], Arlo B. Weil[6]
[1]Frontier Research Institute for Interdisciplinary Sciences, Tohoku University
[2]Department of Earth Science, Tohoku University
[3]Center for Northeast Asian Studies, Tohoku University, 41 Kawauchi, Aoba-ku, Sendai, 980-
8576, Japan. pastor.galan.daniel.a8@tohoku.ac.jp
[4]Dept. of Geology. Faculty of Sciences. University of Salamanca. Plaza de la Merced s/n.
38007, Salamanca (Spain). gabi@usal.es
[5]Geology and Geography Department, Tomsk State University, Lenin Street, 36, Tomsk 634050
Russia
[6]Department of Geology, Bryn Mawr College, PA, USA 19010

## 12   Abstract

13         The collision between Gondwana and Laurussia that formed the latest supercontinent,

Pangea, occurred during Devonian to Early Permian times and resulted in large-scale orogeny
that today transects Europe, northwest Africa and eastern North America. This orogen is
characterized by an 'S' shape corrugated geometry in Iberia. The northern curve of the
corrugation is the well known and studied Cantabrian (or Ibero-Armorican) Orocline and is
convex to the east and towards the hinterland. Largely ignored for decades, the geometry and
kinematics of the southern curvature, known as the Central Iberian curve, are still ambiguous
and hotly debated. Despite the paucity of data, the enigmatic Central Iberian curvature has
inspired a variety of kinematic models that attempt to explain its formation with little consensus.
This paper presents the advances and milestones in our understanding of the geometry and
kinematics of the Central Iberian curve from the last decade, with particular attention to
structural and paleomagnetic studies.

When combined, the currently available datasets suggest that the Central Iberian curve

did not undergo regional differential vertical-axis rotations during or after the latest stages of the
Variscan orogeny, and did not form as the consequence of a single process. Instead, its core is
likely a primary curve (i.e. inherited from previous physiographic features of the crust) whereas
the curvature in areas outside the core are dominated by folding interference during the
Variscan orogeny or more recent Cenozoic (Alpine) tectonics.



## Keywords

**Central Iberia Curve, Variscan orogen, Iberia, Cantabrian Orocline, Curved orogens, Pangea**



## 1 Introduction

Mountain belt systems are the most striking product of plate tectonics. In addition to their
astonishing visual effect, marking the locations where ancient and modern plates collided,
orogenic belts often preserve a variety of rocks that have the potential to illuminate the entirety
of the systems pre- and syn-orogenic history. One of the most striking characteristics of the
majority of Earth's orogens are their curvature in plan-view (e.g. van der Voo, 2004; Marshak,
2004; Rosenbaum, 2014). The degree of orogenic curvature may range from a few degrees of
deflection in structural trend (e.g. Kopet Dag, Iran), to 180° of arc curvature (e.g. Kazakhstan arc
and the Carpathians). The kinematics, structural and geodynamic implications of these systems
are as varied as their geometries (Marshak, 2004; Weil and Sussman, 2004; Johnston et al.,
2013). For example, some orogenic curvatures are hypothesized to be the consequence of
physiographic features of the basement that pre-date orogen formation, such as irregular basin
architectures or plate margin salients and recesses (e.g. Jura mountains, Hindle et al., 2000),
which then control the growth geometry of the ensuing orogen. These systems are known as
primary arcs and reflect pre-orogenic geometries and show no significant or systematic vertical-
axis rotations along their structural length. On the other hand, oroclines, as classically defined
by Carey in 1956, involve systematic differential vertical-axis rotations subsequent to initial
orogenic shortening: different sectors of an orogen rotate with variable magnitudes or in
opposite directions (e.g. Li et al., 2012). Rotations in Oroclines may occur at a range of scales,
from thrust emplacement at upper crustal levels (e.g. Izquierdo-Llavall et al., 2018), up to a
lithospheric-scale vertical-axis folding (e.g. Li et al., 2018). They can occur as single curves (e.g.
Maffione et al., 2009), coupled curves (Johnston, 2001), or in trains of curves (Li and
Rosenbaum, 2014). Oroclines can form during the main orogenic building event, known as
progressive oroclines (Johnston et al., 2013; e.g., the Wyoming salient, Yonkee and Weil, 2010,
and Weil et al., 2010) or in a subsequent tectonic pulse, so-called secondary oroclines (Weil
and Sussman, 2004). Understanding the kinematics and mechanisms of curvature formation in
mountain belts is a critical step to understanding orogenesis in 4D and to evaluate their
geodynamic consequences and paleogeographic implications.
The Variscan-Alleghanian orogeny resulted in the suturing of Gondwana and Laurussia
during Devonian-Carboniferous times, and ultimately led to the formation of the supercontinent,
Pangea. This long and sinuous orogen runs for >8000 km along strike and is ca. 1000 km wide,
transecting across Europe, to northwest Africa and into eastern North America. The final stages
of Pangea amalgamation (e.g. Nance et al., 2010) modified the Western Europe sector of the



belt into its characteristic sinuous shape, which today traces at least one, and perhaps four arcs
from Poland to Brittany, and then across the Bay of Biscay (Cantabrian Sea) into Iberia, where
the system is today truncated by the Betic Alpine orogeny in southeast Iberia (Fig. 1; e.g. Weil
et al., 2013). The southern truncation of the Variscan in Europe hinders a precise correlation
with equivalent age outcrops in NW Africa.

Within the Iberian Peninsula, the orogen is characterized by two large-scale curves (Fig.

2): (1) to the north is the well studied and nearly 180˚ secondary orocline, the Cantabrian (a.k.a.
Ibero-Armorican) Orocline, which buckled a segment of the Variscan belt from ~315 to ~290 Ma
(e.g. Weil et al., 2019 and references therein); and (2) to the south is a curve with disputed
magnitude and kinematics, and is usually referred to as the Central Iberian curve/orocline or
Castillian bend (Martínez-Catalán et al., 2015). Though there remains tremendous uncertainty
on the geometry and kinematics of the Central Iberia curve, multiple hypotheses exist as to its
nature, and disagreements continue on its importance in the tectonic evolution of Europe during
the waning stages of Paleozoic global supercontinent construction. The diversity of author's
interpretations of the Central Iberian curve range from a nonexistent structure (Dias et al.,
2016), to being one of the most important pieces to our understand of the late Carboniferous
and Permian geodynamics of the Iberian Variscan system (e.g. Martínez-Catalán et al., 2011;

2014).

This paper reviews the most recent advances on the geometry and kinematics of the

Central Iberia curve, synthesizing what we know and what we don't, and ending with a
discussion of the main unsolved issues. We hope that this paper fosters novel studies that will
lead to a better understanding of when and which mechanisms acted in the aftermath of the
Variscan-Alleghanian orogeny.

## 2 The long and winding orogen

The Variscan (Europe-NW Africa)-Alleghanian (North America) orogeny is a continental-

scale tectonic system (1000 km wide and 8000 km long) that sutured Gondwana and Laurussia
together, forming the supercontinent Pangea (e.g. Domeier and Torsvik, 2014; Edel et al., 2018;
Pastor-Galán et al., 2019a). The fragments of this system are now dispersed over three
continents, Europe, Africa and North America due to the Mesozoic break-up of Pangea (Buiter
and Torsvik, 2014; Keppie, 2015). This orogen formed as a consequence of a long and
protracted tectonic history that involved several different events, from initial convergence (ca.
420 Ma; e.g. Franke et al., 2017), to the consumption of multiple putative oceanic tracts and/or
basins that existed between Gondwana and Laurussia (ca. 280 Ma; e.g. Kirsch et al., 2012).



The Variscan-Alleghanian orogen itself represents the closing of at least one major ocean, the
Rheic (e.g. Nance et al., 2010), whose axial ridge likely failed or subducted at ca. 395 Ma along
its paleo-northern margin (e.g. Woodcock et al., 2007; Gutiérrez-Alonso et al., 2008a). Perhaps
the orogeny involved other large oceans (Stampfli and Borel, 2002; Franke et al., 2017; 2019),
but most surely involved several minor seaways and basins that existed between Gondwana,
Laurussia, and several intervening micro-continents (e.g. Azor et al., 2008, Dallmeyer et al.,
1997; Kroner and Romer, 2013; Díez-Fernández et al., 2016; Pérez-Cáceres et al., 2017). The
final continent-continent collision began after closure of all oceans and intervening seaways.
The commencement of this deformation was diachronistic and became progressively younger
westwards (in present-day coordinates): with Devonian continent-continent collisions along the
eastern boundary, progressing to earliest Permian ages in the westernmost sector (McWilliams
et al., 2013; Chopin et al., 2014; López-Carmona et al., 2014; Franke et al., 2017).
The present-day geometry of the Variscan-Alleghanian systems has a contorted trace
(Fig. 1). In Europe, from east to west, the trend starts with a prominent curve around the
Bohemian massif (e.g. Tait et al., 1996), followed by a deflection in the Ardennes-Bravant (e.g.
Zegers et al., 2003). In Brittany the outer curvature of the Cantabrian or Ibero-Armorican
orocline begins (e.g. van der Voo et al., 1997), and wraps nearly 180° around across the Bay of
Biscay as it turns in NW Iberia. The Central Iberian curve marks the final concave to the west
curve (in present-day coordinates) and is the focus of this paper (e.g. Aerden, 2004; Martínez
Catalán, 2011; Shaw et al., 2012). The orogen continues in North America where, from north to
south, it has salients and recesses that undulate back and forth from Atlantic Maritime Canada
(e.g. O'Brien, 2012) down along the Pennsylvanian and Alabama curves (e.g. Thomas, 1977).
Interpretation on the origin of these curvatures varies widely. The curvatures in North
America are argued to be the result of a preexisting irregular margin of Laurentia due to the
break-up of the Rodinia supercontinent, which resulted in the formation of orogenic salients and
recesses during subsequent Appalachian collision (e.g. Rankin, 1976; Thomas, 1977, 2004). In
this case, vertical-axis rotations affected only the upper crustal levels during orogenesis (e.g.
Marshak, 1988; Bayona et al., 2003; Hnat and van der Pluijm, 2011). In Europe, the Bohemian
and Ardennes-Bravant massif curvatures have poor kinematic constraints. In the Bohemian
Massif, some suggest secondary rotations that formed an orocline (Tait et al., 1996), while
others suggest little to no vertical-axis rotations and a primary arc (Chopin et al., 2012). The
Ardennes-Bravant Massif record some vertical-axis-rotations (e.g. Molina-Garza and
Zeijderveld, 1996), but it is unclear if these are a response to progresive or secondary oroclinal
bending, or whether rotations only affected the upper crust. The most outstanding example of





Variscan-Alleghanian orogen curvature is exposed in the Iberian Massif, with the Cantabrian
Orocline and the coupled curvature of Central Iberia.

## 136    2.1 Two of us: The Variscan orogen in Iberia

The western half of the Iberian Peninsula constitutes the Iberian massif, one of the

largest exposures of the Variscan orogen and the only place that contains an almost continuous
cross section of the orogen (Fig. 2; e.g. Lotze 1945, Julivert 1974, Pérez-Estaún et al., 1991;
Ayarza et al., 1998; Simancas et al., 2003; Ribeiro et al. 2007, Martínez Catalán et al., 2014,
2019). The majority of the Iberian Massif contains Gondwanan affinity rocks (e.g. Murphy et al.,
2008; Pastor-Galán et al., 2013a; Gutiérrez-Marco et al., 2017) and likely represents a proximal
piece of the Gondwana margin until its final amalgamation with Pangea (e.g. Pastor-Galán et
al., 2013b). Owing to the stratigraphic, structural and petrological styles, the Iberian Massif has
been traditionally divided into six tectonostratigraphic zones (Fig. 2; Lozte, 1945; Julivert, 1971):
(1) Cantabrian Zone represents a Gondwanan thin-skinned foreland fold-and-thrust belt. It has
overall low-grade internal deformation and metamorphism, and represents shortening that
occurred during Mississippian times (e.g. Marcos and Pulgar, 1982; Pérez Estaún et al., 1988;
Gutiérrez-Alonso 1996; Alonso et al., 2009; Pastor-Galán et al., 2009; 2013b). (2) The West-
Asturian Leonese Zone represents a metamorphic fold-and-thrust belt with barrovian
metamorphism that collapsed coevally with thrust emplacement onto the Cantabrian Zone (e.g.
Martínez-Catalán et al., 1992; Alcock et al., 2009; Martínez-Catalán et al., 2014). (3) The
Central Iberian Zone represents the Gondwanan hinterland with Barrovian and Buchan
metamorphism and is intruded by igneous rocks of various ages (e.g. Macaya et al., 1991; Díez
Balda, 1995; Gutiérrez-Alonso et al., 2018). (4) The Ossa-Morena Zone represents the most
distal zone of the Gondwana platform, and is characterized by a metamorphic fold-and-thrust
belt with dominantly sinistral displacement (e.g. Robardet and Gutiérrez-Marco, 2004; Quesada,
2006). (5) The Galicia-Tras-os-Montes Zone represents a far travelled allochthonous terrane
that contains high pressure units and relics of oceanic-like crust (e.g. López-Carmona et al.,
2014; Martínez-Catalán et al., 2019). (6) The South Portuguese Zone represents a foreland
fold-and-thrust belt with little internal deformation and metamorphism with Avalonian affinity and
a strong left-lateral component of shear (e.g. Pereira et al., 2012; Pérez-Cáceres et al., 2016;
Oliveira et al., 2019). Geographically, the external zones of the Gondwana margin are nested to
the north into the core of the Cantabrian Orocline, whereas the hinterland zones are to the west
and center of the massif (Fig. 2; e.g. Díaz Balda, 1995; Azor et al., 2019). The southwestern-
most extent of Iberia contains a putative suture of the Rheic ocean, as well as a piece of the





Laurussian margin fold-and thrust belt, today preserved in the South Portuguese Zone (e.g.
Pereira et al., 2012, 2017; Oliveira et al., 2019).
The Gondwanan authocton stratigraphy (Cantabrian, West Asturian-Leonese, Central
Iberian and Ossa Morena Zones) consist of a Neoproterozoic arc and back-arc basin (e.g.
Fernández-suárez et al., 2014), which evolved to a rift-to-drift Cambrian to Early Ordovician
sequence and then to an Ordovician to Late Devonian passive margin basin sequence (e.g.
Sánchez-García et al., 2019; Gutiérrez-Marco et al., 2019; Gutiérrez-Alonso et al., submitted).
Overall the system transitioned from a relatively isolated Early Cambrian continental, to a
restricted marine basin, to development of an open marine platform that was locally punctuated
by magmatism (e.g. Gutiérrez-Alonso et al., 2008b; Palero-Fernández, 2015). The Ossa
Morena zone represents the outermost platform, followed by an intermediate platform
characterized by an asymmetric horst (Central Iberian Zone) and graben (West-Asturian
Leonese Zone), which ends in the innermost shelf environment of the Cantabrian zone (Fig. 3;
e.g. Gutiérrez-Marco et al., 2019). The differences between the West Asturian-Leonese and
Central Iberian Zone are mainly deeper vs. shallower sedimentary facies (respectively) and a
local Lower Ordovician unconformity in the Central Iberian Zone (Toledanian, e.g. Álvaro et al.,
2018) that places Lower Ordovician strata atop pre-Cambrian to Cambrian rocks (Fig. 3; e.g.
Gutiérrez-Marco et al., 2019). The Central Iberian Zone is divided into two domains: (1) The Ollo
de Sapo domain, which contains abundant Lower Ordovician calc-alkaline magmatism (e.g.
Díez Montes, 2006; Gutiérrez-Marco et al., 2019); and (2) the 'Schistose–greywacke Domain'
characterized by a predominance of outcrops of Neoproterozoic to Lower Cambrian
sedimentary rocks (e.g. Gutiérrez-Marco et al., 2019 and references therein).
The Galicia Tras-os-Montes Zone (Farias et al., 1987) is a complex structural stack
including a basal schistose unit (Parautochthon; Dias da Silva et al., in press) structurally
overlain by mafic rocks with an oceanic-like signature and other far-traveled rocks under high-
pressure metamorphism (e.g. López-Carmona et al., 2014; Martínez-Catalán et al., 2019). The
oceanic rocks of this zone are classically interpreted as a Rheic Ocean suture (e.g. Martínez
Catalán et al., 2009). Recent interpretations support its origin as a minor oceanic basin or
seaway within the realm of Gondwana (e.g. Pin et al., 2002; Arenas et al., 2016).
The South Portuguese Zone constitutes the Laurussian foreland fold-and-thrust belt in
the Iberian Variscides (e.g. Pereira et al., 2012; Pérez-Cáceres et al., 2017). It contains three
units: (1) the Pulo de Lobo, a low grade metamorphic accretionary prism with clastic
sedimentary rocks and basalts with MORB signature (e.g. Azor et al., 2019; Pérez-Cáceres et
al., this volume); (2) The Iberian Pyrite Belt, which is a world class volcanogenic massive sulfide





deposit formed between 390 and 330 Ma (e.g. Oliveira et al., 2019a; 2019b); and (3) the Baixo
Alentejo Flysch, which is located to the southwest and is a syn-orogenic composite turbiditic
sequence with ages from ~330 to ~310 Ma (Oliveira et al., 2019b). The boundary between the
South Portuguese and Ossa Morena zones is a sinistral shear zone (so-called Beja-Acebuches,
Quesada and Dallmeyer., 1994; Pérez-Cáceres et al., 2016) that contains a strongly deformed
amphibolitic belt with oceanic affinity (Munha et al., 1986; Munha, 1989; Quesada et al., 2019).
This belt potentially represents dismembered relics of the Rheic ocean and/or a subsidiary
seaway that opened during a Variscan transtension event in SW Iberia (e.g. Pérez-Cáceres et
al., 2015; Quesada et al., 2019).

Finally, Paleozoic rocks occur sporadically within the Alpine Betic chain. Their lithological

monotony, paucity of fossils, and the intensity of deformation and metamorphism during Alpine
orogeny, make recognizing the original features of the different successions challenging (e.g.
Martín-Algarra et al., 2019). Some faunal and detrital zircon studies suggest that the Paleozoic
outcrops in the Betics may be similar to the most seaward realms of the Gondwanan platform
(i.e., the Cantabrian Zone; e.g. Rodríguez-Cañero et al., 2018; Jabaloy-Sánchez et al., 2018).
Following the latest plate reconstructions of the Mediterranean during Meso-Cenozoic times, the
Paleozoic units of the Betic-Rif chain may have been located proximal to the present-day
position of the Balearic Islands (van Hinsbergen et al., 2020).

The Variscan orogen in Iberia shows multiple deformation, metamorphic, and magmatic

events (e.g. Martínez-Catalán et al., 2014; Azor et al., 2019; Fig. 2) that evolved diachronously
from the suture towards the external zones (Dalmeyer et al., 1997): (1) An initial continent-
continent collision began ca. 370-365 Ma, which produced high pressure metamorphism (e.g.
Lopez-Carmona et al. 2014). (2) Between 360 and 330 Ma a protracted shortening phase
occurred, frequently divided into main phases C1 and C2, that were accompanied by Barrovian
type metamorphism (e.g. Dias da Silva et al., in press) and plutonism at ~340 Ma (e.g.
Gutiérrez-Alonso et al., 2018). (3) An extensional collapse event, so-called E1, occurred at
~333-317 Ma, which formed core-complexes and granitic domes in the Central Iberian and
West Asturian-Leonese zones (Fig. 2C; e.g. Alcock et al., 2009; Díez-Fernández and Pereira,
2016; López-Moro et al., 2018). This event is coeval and genetically linked to the formation of
the foreland fold-and-thrust-belt of the Cantabrian Zone (e.g. Gutiérrez-Alonso, 1996). (4) A late
Carboniferous shortening event (C3) occurred ca. 315-290 Ma and is argued to have resulted in
the formation of the Cantabrian Orocline and was accompanied by the intrusion of mantle
derived granitoids (Fing. 2C; e.g. Gutiérrez-Alonso et al., 2011a, 2011b; Pastor-Galán et al.,
2012a). (5) A final early Permian extensional event (E2), mostly found in the Central Iberian





Zone, resulted in the formation of core complexes and regional doming (Dias da Silva et al., in
press). (6) A final shortening event (C4), possibly coeval with E2, resulted in widespread brittle
deformation (e.g. Azor et al., 2019; Fernández-Lozano et al., 2019).

In SW Iberia, the aforementioned Variscan deformation events are characterized by a

dominant sinistral component, which contrasts with the general dextral component recognized in
most other regions of the orogen (e.g. Martínez Catalán et al., 2011; Gutiérrez-Alonso et al.,
2015). Early collisional structures (C1) formed NE-vergent recumbent folds in the southernmost
Central Iberian Zone and SW-vergent folds and thrusts in the Ossa Morena and South
Portuguese zones. This phase continued with a transtensional event that heterogeneously
extended the continental lithosphere (e.g. Pérez-Cáceres et al. 2015). Coevally, an important
extension-related magmatic event happened, perhaps assisted by a plume-type mantle
(Simancas et al. 2006) or a slab break-off (Pin et al. 2008). After this transtensional event,
significant left-lateral transpression occurred forming the extensive shear zones to the north and
south of Ossa Morena Zone (Fig. 2B), which accommodated the majority of the transcurrent
motion. However, left-lateral displacements are observed all along the Ossa Morena and South
Portuguese zones. Pérez-Cáceres et al. (2016) estimated over 1000 km of collisional
convergence in SW Iberia, most of which corresponds with left-lateral displacements parallel to
terrane boundaries.

## 253 3 Synthesis on the Geometry and Kinematics of the Cantabrian
## 254 Orocline

Understanding the geometry, kinematic evolution and mechanics of curved mountain

systems is crucial to developing paleogeographic and tectonic reconstructions (e.g. Marshak,
2004; Van der Voo, 2004; Li et al., 2012; van Hinsbergen et al., 2020). Introduced by Carey
(1955 p.257), an orocline (from Greek ορος, mountain, and κλινο, bend) is "...an orogenic
system, which has been flexed in plan to a horse-shoe or elbow shape." Although sometimes
used in the literature as a geometric description of any orogenic curvature, herein orocline is
strictly used as a the term for map-scale bends that underwent vertical-axis rotations (Weil and
Sussman, 2004; Johnston et al., 2013; Pastor-Galán et al., 2017a). The kinematic classification
of curved mountain belts (Weil and Susman, 2004; Johnston et al, 2013) distinguishes two end
members: (1) Primary orogenic curves, which describe those systems in which curvature is a
primary feature of the orogen and formed without significant or systematic vertical-axis rotations,
and (2) Secondary oroclines, where orogenic curvature was acquired due to vertical-axis
rotations subsequent to primary orogenic building. Those systems whose curvature is the





product of vertical-axis rotation during the primary orogenic pulse and/or only a portion of the
observed curvature is secondary are progressive oroclines.
The orocline test (or strike test), evaluates the relationship between changes in regional
structural trend (relative to a reference trend for an orogen) and the orientations of a given
geologic fabric element or magnetization (relative to a reference direction). In terms of
evaluating developmental kinematics, the most relevant geologic marker is paleomagnetic
declination, which can be used to quantitatively evaluate total and systematic rotations as a
function of along-strike variability. Once acquired, data is plotted on Cartesian coordinate axes
with the strike (S) of the orogen (relative to a reference) along the horizontal axis, and the fabric
azimuth (F, relative to a reference) along the vertical axis. The test originally used a basic least-
squares (OLS) regression (Schwartz and Van der Voo, 1983) to estimate the slope (coded m in
formulas), ideally between 0 and 1, which then is interpreted with respect to vertical-axis
kinematics. More recently, Yonkee and Weil (2010b) and Pastor-Galán et al. (2017a) introduced
more robust statistics to estimate the slope and its uncertainty, considering and propagating
errors of the input data. Primary orogenic bends show no change of paleomagnetic declination
orientations with varying structural trend, and therefore the slope is expected to be 0. In
progressive oroclines, the declination variation records some fraction of the total observed
orogenic strike variability, and thus the slope would range between 0 and 1, depending on the
amount of primary curvature. Secondary oroclines are those in which the paleomagnetic vectors
record 100% of the rotation, yielding slopes of 1, meaning that the orogenic system must have
started as a roughly linear system that then underwent secondary vertical-axis rotations until its
present-day curvature was acquired. The slope obtained with the orocline test can only be
confidently interpreted when the chronology of fabric formation is well known.
The trend of the Variscan belt in Iberia follows a sinuous "S" shape that is especially
prominent in the northwest region of the Iberian Peninsula, and then becomes more subtle due
to the predominance of younger cover sequences in the central and eastern regions of the
peninsula (Fig. 1 and 2). This dramatic geometry has stimulated a century long scientific debate
as to its origin (e.g. Suess, 1892; Staub, 1926; Martínez Catalán et al., 2015). To the north and
convex to the west is the Cantabrian Orocline, and to the center-south and convex to the east is
the Central Iberian curve. The overall trend of the Cantabrian Orocline starts in Brittany (France)
and southern England and then curves through the Bay of Biscay and then south into central
north Iberia (Fig. 1, 2 and 4). The Cantabrian Orocline (also known as Ibero-Armorican Orocline/
Arc, Asturian Arc or Cantabrian-Asturias Arc) is arguably the first curved orogen that was
scientifically described, recognized by the change in structural trend of mapped thrusts and fold





axes (Schultz, 1858, Barrois, 1882, Suess, 1892). The Cantabrian Orocline traces an arc with a

curvature close to 180˚ within the central Cantabrian Zone (the Gondwanan foreland in Iberia,

fig. 2), and opens to approximately 150˚ as one moves to the outer arc reaches (Fig. 1). At the

crustal-scale, the Cantabrian Orocline represents a first order vertical-axis buckle fold in plan-

view that refolds pre-existing Variscan structures (e.g. Julivert and Marcos, 1973; Weil et al.,

2001). The inner arc of the orocline, or the Cantabrian Zone is characterized by tectonic

transport towards the core of the orocline, i.e., the orocline has a contractional core, where low

finite strain values and locally developed cleavage occur (Pérez-Estaún et al., 1988; Gutiérrez-

Alonso, 1996; Pastor-Galán et al., 2009). Within the inner core a variety of structures record

non-coaxial strain, which produced complex interference folds and rotated thrust sheets (e.g.

Julivert and Marcos, 1973; Julivert and Arboleya, 1984; Pérez-Estaún et al, 1988; Aller and

Gallastegui, 1995: Weil, 2006, 2013; Pastor-Galán et al., 2012b; Shaw et al., 2015; 2016a; Del

Greco et al., 2016). In contrast, the outer arc shows a ca. 150˚ interlimb angle vertical-axis fold

that was accommodated by significant shearing, both dextral and, in lesser amounts, sinistral

penecontemporaneous to vertical-axis rotation (Gutiérrez-Alonso et al., 2015). Weil et al. (2013,

2019) extensively review the geometry of the Cantabrian Orocline.

All kinematic data studied so far support a model in which the Cantabrian Orocline

formed due to secondary vertical-axis rotation in a period of time younger than 315 Ma and

older than 290 Ma. Overall, the southern limb of the orocline rotated counterclockwise (CCW)

and the northern limb clockwise (CW; Fig. 4). Orocline formation happened subsequent to the

main shortening phases of the orogen (C1 and C2) and late-stage orogenic collapse (E1), and

therefore, it is an ideal example of a secondary orocline in the strictest sense. Development of

the Cantabrian Orocline requires the existence of a roughly linear orogenic belt during early

Variscan closure of the Rheic Ocean (with a roughly N-S orientation in present-day

coordinates), which was subsequently bent in plan-view into an orocline during late stages of

Pangea amalgamation. Such interpretation is grounded in paleomagnetic studies (e.g. Hirt et

al., 1992; Parés et al. 1994; Stewart, 1995; van de Voo et al., 1997; Weil, 2006; Weil et al.,

2000; 2001; 2010), along with important contributions from structural (e.g. Gutiérrez-Alonso

1992; Kollmeier et al., 2000; Merino-Tomé et al., 2009; Pastor-Galán et al., 2011; 2014; Shaw et

al., 2015) and geochronological studies (e.g., Tohver et al., 2008; Gutiérrez-Alonso et al., 2015).

Weil et al. (2013) provides a comprehensive review on the kinematic constraints, updated in

2017a by Pastor-Galán et al., and in 2019 by Weil et al.



## 4 The intriguing geometry of the Central Iberian curve


The more southern Central Iberian curve has a similar magnitude, but opposite
curvature compared to the Cantabrian Orocline (Fig. 1 and 2B). This structure has been referred
to as the Central Iberian curve, arc, bend or orocline. In this paper we use 'Central Iberian
curve'. The other aforementioned terms involve still unknown parameters or are misleading:
orocline imply kinematics (Weil and Sussman, 2004); bend refers to a mechanism of formation
(e.g. Fossen, 2016); and arc could be ambiguous, since the term is commonly used for volcanic
chains. This curvature was first described by Staub (1926) and was termed the Castilian bend.
Continental drift pioneers paid some attention to Staub's description (e.g. Holmes, 1929; Du
Toit, 1937), but the curved structure remained largely ignored for multiple decades (e.g.
Martínez Catalán et al., 2015). The hypothesis of a large-scale curvature in Central Iberia made
a comeback at the beginning of the 21st century with a study of Variscan porphyroblast
kinematics across Iberia by Aerden in 2004. Since then, several attempts to unveil its geometry
and kinematics have been made with contrasting results.
The elusive nature of the Central Iberian curve resides in the poor exposure of its
putative hinge (Fig. 2). The hinge of the Cantabrian orocline crops out extensively and the
changes in thrust and fold axes trend are observable at high-resolution from aerial photographs
and are readily mapped using outcrop-scale observations. In contrast, the alleged hinge of the
Central Iberian curve is largely covered by Mesozoic and Cenozoic basins (Fig. 2). The
curvature is most recognizable at the boundary between the Galicia-Tras os Montes and Central
Iberian zones (Fig. 2A; Aereden, 2004; Martínez Catalán, 2012). The thrust fault that bounds
those areas traces close to a 180˚ of curvature and marks the emplacement of the most distal
units. Before the revival of Staub's curved geometry along the entire Central Iberian Zone, there
were several attempts to explain the curved shape of the Galicia Tras-os-Montes Zone. Some
consider the Galicia Tras-os-Montes Zone a block that escaped during an early Variscan (C1)
non-cylindrical collision, forming a extrusion wedge towards the areas undergoing lesser
amount of shortening (Martínez-Catalán, 1990, Dias da Silva, 2015; in press); or alternatively a
klippe of a larger allochthonous thrust sheet, product of an interference pattern between C2, E1
and C3 structures (e.g. Ries and Shackleton, 1971; Martínez Catalán et al., 2002; Rubio
Pascual et al., 2013; Díez-Fernández et al., 2015).
In addition to the Galicia Tras-os-Montes Zone, other areas showing a certain degree of
curvature are to the E and SE of the Central Iberian Zone. There, an approximately 20° change
in strike of the Iberian ranges (NE Iberia, Fig. 2A) is observed, which represents the only known
outcrop of the hinge of the Central Iberian curve's outer arc. The rest of the curvature has been



deduced with indirect observations leading to three competing geometric proposals for the
Central Iberia curve (Fig. 2B). The main arguments used to constrain the geometry of the
Central Iberian curve are: (1) the geometry of Galicia Tras-os-Montes folds and the orientation
of observed garnet inclusion trails (Aerden, 2004; Fig. 2B-1); (2) aligned aeromagnetic
anomalies and fold trends in the Iberian ranges and the E-SE Central Iberian Zone (Martínez-
Catalán, 2012; Fig. 2A and 2B-2) and; (3) the regional distribution of paleocurrents recorded in
Ordovician quartzites (Shaw et al., 2012; Fig. 2B-3 and 3). All proposed geometries share two
features: (1) The curvature runs parallel to the Central Iberian Zone, and is located in the
center-west of Iberia, and (2) all place the Galicia Tras-os-Montes Zone in the core of the curve
with the curves axial trace cross-cutting the Morais Complex, a set of mafic and ultramafic rocks
that is roughly circular in shape (Fig. 2B; Dias da Silva et al., in press).

Aerden (2004) compared the orientation of inclusions in metamorphic porphyroblasts
across the Variscan allochthonous terranes of the NW Iberian Massif, and found that inclusion
trails maintain a constant north–south orientation. Comparing such results with the trend of the
Variscan fold axes in the central Iberian Zone (Fig. 2A) and a daring interpretation of the
aeromagnetic anomalies of the Iberian Peninsula (Fig. 5A), Aerden suggested a geometry in
which the Central Iberian curve was more prominent in the outer arc than in the inner arc (Fig.
2B-1). In Aerden's view the geometry of the Galicia Tras-os Montes Zone does not represent a
large-scale curvature, but rather the original shape of the nappe, perhaps re-tightened during
C3 deformation. In contrast, the Iberian Ranges and the SE Central Iberian Zone represent the
more curved sector (Fig. 2B-1). In the model of Aerden (2004), the Ossa Morena and South
Portuguese Zones are not part of the Central Iberian curvature.

Martínez-Catalán (2012) reinterpreted Aerden's analysis of aeromagnetic map data (Fig.
5A) and the interpretive structural trends of C1-C2 fold axes from Central Iberian Zone
structures (Fig. 2A). In Martínez-Catalán's model, the Central Iberian curvature is a symmetric
arcuate shape in which orogen trend changes equally in the inner and outer arc, and is
comparable in size to the Cantabrian Orocline, but with opposite curvature and less shortening.
This geometric model also excludes the Ossa Morena and the South Portuguese Zones as
elements involved in the formation of the curvature (Fig. 2B-2).

Finally, Shaw et al. (2012) studied the orientation of paleocurrents in Ordovician
Armorican Quartzite (e.g. Aramburu, 2002), which is one of the most prominent rocks exposed
in Iberia (Fig. 3). The authors found that paleocurrents fanned outward with respect to the
Cantabrian Orocline curve and are approximately perpendicular to the structural trend
throughout the peninsula (Fig. 3). Shaw et al. (2012) assumed that the direction and sense of





paleocurrents were parallel throughout all zones, and concluded that the Central Iberia curve is
a 'S' shape isoclinal structure similar in magnitude to the Cantabrian Orocline (Fig. 2B-3). It is
unclear from the Shaw et al. (2012) model the involvement of the Ossa Morena and South
Portuguese Zones in the overall curve (if any), nor the prospective location of the external zones
of the orogen (Cantabrian Zone) with respect to the overall curvature.

# 5 Move over once, move over twice: Kinematic constraints

Late Variscan kinematic data (315-290 Ma; C3, E2, C4 phases) in the Central Iberian
curve were scarce prior to revival of Staub's Central Iberian curve (e.g. Vergés, 1983; Julivert et
al., 1983; Parés and van der Voo, 1992). More recently, a wealth of studies have been
published on the kinematics of forming the Central Iberian curve (Fig. 2B), which are reviewed
below.

## 5.1 Structural Geology and Geochronology

Curved orogens that result from differential vertical-axis rotations develop remarkable
structures within their hinges where compressive and extensive radial structures often develop
in combination with tangential shear structures (e.g. Li et al., 2012; Eichelberger and McQuarrie,
2015). With the re-emergence of the Central Iberian curve debate, several studies have re-
evaluated the well-documented structures from the Central Iberian Zone to constrain the origin
and kinematics of curvature. The majority of studies focused on the hinge zone of the curve in
the area surrounding Galicia Tras-os-Montes (e.g. Dias da Silva et al., 2014; Jacques et al.,
2018a), but some explored more outer-arc areas (e.g. Palero-Fernández et al., 2015; Gutiérrez-
Alonso et al., 2015). The following paragraphs synthesize the findings of new field, structural,
and geochronological analyses from around the hinge of the Central Iberian curve and its
surrounding regions. The reviewed studies identify several deformation events that are linked to
regional Variscan deformation phases (Fig. 2A).

1.  An early generation of upright to overturned cylindrical folds with an associated axial
    planar cleavage (C1). The C1 fold axes plunge variably from horizontal to nearly vertical
    (e.g. Jacques et al., 2018a, 2018b). The original trend of the fold axes was parallel to the
    orogen (e.g. Pastor-Galán et al., 2019b), however interference with younger deformation
    events has created complicated geometries (e.g. Díez Fernández et al., 2013; Palero-
    Fernández et al., 2015). The emplacement of the allochthonous units of Galicia Tras-os-
    Montes zone (commonly referred as C2) is closely associated with development of C1
    folds, but is restricted to shear zones located along the boundary between the latter and



the Central Iberian Zone. This phase includes orogen-parallel emplacement of the
allochthonous Galicia Tras-Os Montes units and its associated thrusts (Fig. 2A). The
non-coaxial nature of the emplacement of this allochthonous nappe produced folding
interference and local vertical-axis rotations (Dias da Silva et al., in press). Prograde
Barrovian metamorphism (known as M1) reached its pressure peak at the end of C2
(Rubio Pascual et al., 2013).
2.  After C1 and C2, the resulting thickened crust gravitationally collapsed (Macaya et al.,

1991; Escuder Viruete et al., 1994; Díaz-Balda et al., 1995; Díez-Montes, 2010). This

gravitational collapse (phase E1) formed gneiss-dome core complexes between 330 and

317 Ma (e.g. Díez Fernández and Pereira, 2016) especially at the core of the Central

Iberian curve (Fig. 2C; e.g. Martínez-Catalán, 2012). This phase formed large

subhorizontal extensional detachments that exhumed to depths of the middle crust (e.g.

Rubio-Pascual et al., 2013; Dias da Silva et al., in press). General decompression

produced a Buchan-type metamorphic event (M2; e.g. Rubio-Pascual et al., 2016, Solís-

Alulima et al., 2019) and widespread anatectic melting (e.g. López-Moro et al., 2018;

Pereira et al., 2018). E1 phase developed a fold system with sub-horizontal axes and a

penetrative subhorizontal cleavage (e.g. Dias da Silva et al., in press). Mapped folding

geometries indicate the deflection of C1 folds into overturned positions within the E1

deformation zones (e.g. Díez Fernández et al., 2013; Díez Fernández and Pereira,

2016; Pastor-Galán et al., 2019b). In addition to large-scale extensional deformation and

Buchan metamorphism, E1 developed a regional dome-and-basin pattern, resulting in

portions of the allochthonous terranes tectonically transported into basins (e.g. Días da

Silva et al., in press).

3.  The structures developed during C1-C2 compression and E1 extension, are re-folded by

a younger shortening phase (C3; syn-Cantabrian Orocline). C3 formed upright open

folds and conjugate sub-vertical shear zones (e.g. Gutiérrez-Alonso et al., 2015; Díez

Fernández and Pereira, 2017; Dias da Silva et al., in press). C3 was coeval with regional

retrograde metamorphism (M3) and with intrusion of mantle derived granitoids (Fing. 2C;

e.g. Gutiérrez-Alonso et al., 2011a), surrounded by contact metamorphic aureoles (e.g.

Yenes et al., 1999). The age of the C3 event ranges from 315 and 290 (e.g. Jacques et

al., 2018a), concomitant with the formation of the Cantabrian Orocline (e.g. Pastor-Galán

et al., 2015a). Ductile deformation, including folding with axial planar cleavage (e.g. Dias

da Silva et al., 2014; Pastor-Galán et al., 2019b) and shear zones, occurred at the early

stages of C3 (315-305 Ma; Gutiérrez-Alonso et al., 2015; Díez-Fernández and Pereira,



2017; Jacques et al., 2018b) followed by brittle deformation that formed cross-joint sets
and vein swarms with Sn-W mineralizations (Jacques et al., 2018a; 2018b). The
conjugated shear zones, some of them with hundreds of kilometers of displacement, had
activity during the period 315-305 based on direct Ar-Ar dating of the shear zones
(Gutiérrez-Alonso et al., 2015) and cross-cutting relationships with precisely dated
igneous rocks (Díez-Fernández and Pereira, 2017). Note that these shear zones show a
younger age with respect to the sinistral shear zones that bound the Ossa Morena and
South Portuguese zones (340-330 Ma; e.g. Dallmeyer et al., 1993). New studies in the
Central Iberian Zone have determined that several folds, previously interpreted as C1
(e.g. the Tamames-Marofa-Sátão synform) are C3 structures, possibly nucleated within
existing C1-C2 structures (e.g. Dias da Silva et al., 2017; Jacques et al., 2018b). The
remarkable continuity along the Central Iberian Zone of these folds (Fig. 2A), previously
interpreted as C1 (e.g. Díez-Balda et al., 1990; Abalos et al., 2002; Dias and Ribeiro,
1994; Dias et al., 2016), suggest the ubiquity and importance of this deformation phase.
4.  Subsequent to C3 deformation, a brittle shortening event (C4) together with some late

extensional faults occurred across the region (E2; Fig. 2A; Dias and Ribeiro 1991; Dias

et al. 2003; Rubio Pascual et al., 2013; Arango et al., 2013; Fernández-Lozano et al.

2019; Dias da Silva et al., in press). E2 developed core complex-like structures that

further telescoped the M2 metamorphic isograds between the anatectic cores of gneiss

domes and the hanging wall units. This event also favoured sub-horizontal folding and

kink-band generation in the upper structural levels. Post-Variscan shortening structures

in Northern Iberia are characterized by a N-S compressive regime (C4) allowing the

formation of brittle NNE-SSW and NNW-SSE faults and associated sub-vertical and sub-

horizontal widespread kink-bands (e.g. Aller et al., 2020).

## 5.2 Paleomagnetism

Paleomagnetism investigates the record of the Earth's ancient magnetic field as it is
recorded in the rock record. Among other features, rocks can record the orientation of the
magnetic field at the time of magnetization (e.g. Tauxe, 2010). The recorded magnetic vector
can be geometrically defined by two components: inclination, which is a function of the
paleolatitude (being 90˚ at the poles and 0˚ at the equator) at the time of magnetization
acquisition; and declination, which is a measure of the horizontal angular difference between the
recorded magnetic direction and true north, thereby allowing for the quantification of any
vertical-axis rotations if a north reference direction is known for the region of interest at the time





of magnetization acquisition. Paleomagnetism is the best tool to quantify vertical-axis rotations
in orogens due to the independence of the magnetic field from the orogen deformation and
evolution (e.g. Butler, 1998).

Despite its uniqueness to study paleolatitudes and vertical-axis rotations,

paleomagnetism is not flawless. Paleomagnetic data can yield spurious rotations when the local
and regional structures are not properly studied and their geometries and kinematic histories not
adequately corrected for (e.g. Pueyo et al., 2016). In addition, the age of magnetization
acquisition is not necessarily equivalent to the age of the sampled rock. Remagnetizations are
ubiquitous, especially in orogens (Weil and van der Voo, 2002; Pueyo et al., 2007; Huang et al.,
2017). In remagnetized rocks, the primary magnetization is replaced or overprinted due to a set
of geologic processes acting alone or in concert - usually represented by a combination of
thermal or chemical reactions (Jackson, 1990). Nevertheless, remagnetizations can be useful
for interpreting deformation history if the relative timing of the overprint can be established and a
well-constrained reference direction for that age is known (e.g. Weil et al., 2001; Izquierdo-
Llavall et al. 2015; Calvín et al., 2017).

In addition to knowing the structural geology and the timing of magnetization of the

studied rocks, understanding and quantifying local and regional vertical-axis rotations require a
paleomagnetic reference pole for comparison. Permian and Mesozoic paleomagnetic studies in
Iberia indicate that Iberia was a relatively stable plate from at least Guadalupian times (ca. 270
Ma) to the opening of the Bay of Biscay in the Cretaceous (e.g. Gong et al., 2008; Vissers et al.,
2016). Weil et al. (2010) calculated the most modern Early Permian pole for stable Iberia, which
will be used herein as a reference for any vertical-axis rotation analysis (hereafter, eP pole or
eP component). Weil et al.'s Virtual Geomagnetic Pole (VGP) values are Plat = 43.9; Plong =
203.3 and $\alpha_{95}$ = 5.4 and when transform into paleomagnetic components has a ~0˚ inclination
(equatorial) and declinations that range from 150˚ to 160˚ (from NW to SW respectively)
depending on where in Iberia you are referencing. In Fig. 6 (red arrows), a compilation of
declinations that form part of this composite pole and other eP components found in recent
studies are presented.

For the Central Iberian curve, the voluminous paleomagnetic database from the

Cantabrian Orocline can be used to partially constrain its kinematics (e.g. Weil et al., 2013). The
orocline test for the Cantabrian Orocline (fig. 4) quantifies the degree of differential vertical-axis
rotation of variously striking Variscan segments in northern Iberia. If the Central Iberian curve is
a product of vertical-axis rotation, paleomagnetic declinations would bend around the Central
Iberian curve opposite to that of the Cantabrian Orocline. With a well constrained orocline test,
as in the Cantabrian Orocline (Fig. 4), one can use the paleomagnetic strike-test correlation
slope to establish expected declinations for any along-strike portion of the orogen (Pastor-Galán
et al., 2017b).

Before the resurgence of the Central Iberian curve, the only available pre-Permian

paleomagnetic studies to the South and west of the Cantabrian Zone in the Iberian Massif were
focused on the Beja Gabbroic Massif, Portugal (Perroud et al., 1985) and the Almadén syncline
volcanics (Perroud et al., 1991; Pares & Van der Voo, 1992). The study in the Beja area showed
varied inclinations and declinations in the gabbros, and complex overprints elsewhere. Perroud
et al (1985) did not consider any structural correction for the results, assuming the gabbro was
undeformed. Recently, Dias da Silva et al. (2018) showed that the area underwent intense
deformation during the Carboniferous. Therefore interpretation of this dataset is complicated
without knowing the proper structural correction needed to restore the magnetization to its
palinspastic orientation.

Several articles with new paleomagnetic studies around the Central Iberian curve have

been published since 2015 (Fig. 5). In general, these studies have reported a pervasive late
Carboniferous (320 to 300 Ma) (re-)magnetization in sedimentary and igneous rocks (e.g.
Pastor-Galán et al., 2015a; 2017b; Fernández-Lozano et al. 2016), which is largely
penecontemporaneous to the intrusion of E1 extensional granites (López-Moro et al., 2018) and
C3 syn-orocline mantle derived granitoids (Fing. 2C; e.g. Gutiérrez-Alonso et al., 2011a). The
following section describes the magnetizations from oldest to youngest.

Pastor-Galán et al. (2016) sampled for paleomagnetic analyses both E1 extensional

granites (Fig. 2C; ~320 Ma; e.g. López-Moro et al., 2018) from the Tormes and Martinamor
domes, and C3 mantle derived granitoids in the Central System (Fig. 2C; 305-295 Ma; e.g.
Gutiérrez-Alonso et al., 2011a). Both sets of plutons are located around the Galicias Tras-os-
Montes hinge of the Central Iberian curve (Fig. 6-5). The authors found an original component in
E1 grantites supported by a positive reversal test in both domes (Fig. 7). The magnetization has
an inclination (Inc.) = 15˚ (paleolatitude (λ) = -7.6˚) and declination (Dec.) = 81˚ (Fig. 7), which
imply a northward movement of 700 km and a ~70˚ CCW rotation with respect to the C3
granites that showed an eP component (Dec. ~ 150, Inc. ~ 0). Considering the positive reversal
test in E1 granites and the significant difference in inclinations with respect to C3 granitoids (eP
component), a magnetization age of older than 318 Ma was proposed (pre Kiaman superchron,
317 Ma - 267 Ma, e.g. Langereis et al., 2010), which was interpreted as a primary
magnetization. The 70˚ CCW Pennsylvanian rotation recorded in rocks from the Central Iberian
curve hinge zone is in agreement with the expected rotation of the southern limb of the





Cantabrian Orocline (Fig. 4; Weil et al., 2013).

At the putative outer arc of the Central Iberian curve, the Iberian Ranges (Fig. 2),

paleomagnetic and structural studies of Devonian and Permian rocks (Pastor-Galán et al.,
2018) revealed that the eP component from Permian rocks had rotated ~22° CW during the
Cenozoic (Fig. 8; cf. Pastor-Galán et al., 2018). The Permian and Mesozoic rocks from the
Iberian Ranges show a consistent ~22° CW rotation with respect to the Apparent Polar Wander
Path for Iberia (e.g. Pastor-Galán et al. 2018). This rotation likely happened during the Alpine
orogeny, in which the northern area of the Iberian Range underwent more shortening than the
southern part, resulting in a regional CW vertical-axis rotation (Izquierdo-Llavall et al., 2019).
After restoring the Cenozoic rotation, the Devonian rocks show a positive reversal and fold-test
with inclinations that are steeper than expected from the eP component (Dec. = 85.3°, Inc. =
12.7°, λ = -6.4). This component is statistically indistinguishable from that of the E1 granites and
the southern branch of the Cantabrian Zone, showing the same 70° CCW rotation from the time
they were remagnetization (estimated in 318 Ma) to the timing of the eP component (Fig. 8;
Pastor-Galán et al., 2018). Once Cenozoic rotation is corrected for, the structural and
paleomagnetic trends of the Iberian ranges become parallel to those in the southern limb of the
Cantabrian Orocline, ruling out a Variscan or older origin for the outer Central Iberian curve (Fig.

8).

The remaining paleomagnetic works published on Central and SW Iberia rocks all reveal

a ubiquitous late Carboniferous to Early Permian remagnetizations during the Kiaman
superchron (Fernández-Lozano et al., 2016; Pastor-Galán et al., 2015a; 2016; 2017b; Leite
Mendes, in press). The authors of these papers calculated the expected declination for each
site as if they were part of the Cantabrian Orocline (Fig. 9A). All localities where magnetizations
pre-date the formation of the Cantabrian Orocline show the same expected rotations as the
southern limb of the Cantabrian Orocline, regardless of their position within the Central Iberian
curve (to the hinge: Tormes and Martinamor domes, Iberian ranges; to the southern limb:
Almadén syncline and South Portuguese Zone). Other locations, especially limestones from the
Central Iberian Zone, have declinations and inclinations in between the primary 318 Ma
component of the E1 granites and the post-orocline eP component (Fig. 9B). Pastor-Galán et al.
(2015a; 2016) interpreted these results as being caused by a remagnetization that was acquired
during Cantabrian Orocline formation and therefore recorded intermediate steps between the
component of the E1 granites and eP. Those authors suggest that the large amount of syn-
orocline mantle derived granitoids that intruded the Central Iberian Zone (C3 granitoids)
triggered the hinterland remagnetization.





Finally, two previous studies identified an earlier magnetization in the Almadén syncline

region of the SE Central Iberian Zone (Perroud et al., 1991; Pares & Van der Voo, 1992).
However, Leite Mendes et al. (in press) argue that these studies are likely misinterpreted.
Perroud et al. (1991), applied a complicated structural correction restoring a putative plunge of
the regional structural axis to all sites, including those where the syncline axis does not plunge.
Leite Mendes et al. (in press) re-sampled the syncline where its axis is sub-horizontal and
obtained a negative fold test, implying that the magnetization is not primary as previously
interpreted. Their results, however, are similar in orientation to those components published
from previous studies prior to any structural correction (Perroud et al., 1991 and Parés and van
der Voo, 1992).

Two additional studies sampled Laurussian margin sequences that are today adjacent to

the Cantabrian Orocline region (Fig. 10). To the north, the SW area of Ireland preserves a Late
Paleozoic basin filled with Devonian red sandstone and Carboniferous limestone and siltstone,
which was sampled by Pastor-Galán et al. (2015a). To the south is the aforementioned results
from the South Portuguese Zone (Leite Mendes et al., in press). Both areas are interpreted to
have previously been part of the Laurussian continent, on the opposite side of the Rheic Ocean
suture at the time of Variscan collision (Fig. 10; e.g. Pastor-Galán et al., 2015b). In contrast, the
rest of Iberia was part of, or proximal to, Gondwana (e.g. Franke et al., 2017). These
Paleomagnetic results from the Laurussian margin suggest that the rotations involved in the
formation of the Cantabrian Orocline occurred along both sides of the Rheic suture proximal to
both its northern and southern limb (Fig. 10A and B). Pastor-Galán et al. (2015b) hypothesized
a so-called Greater Cantabrian Orocline that would have bent the entire Appalachian/Variscan
orogen around a vertical-axis, affecting at least the continental margins of both Gondwana and
Laurussia.
## 5.3 The implications of not being a secondary orocline

The most relevant new data regarding the kinematics of the Central Iberian curve is the

paleomagnetic study from the Iberian Ranges (Calvín et al., 2014; Pastor-Galán et al., 2018).
These results confirm that the present-day variation in trend of the tectonostratigraphic units,
generally attributed to Variscan tectonics (e.g. Weil et al., 2013; Shaw et al., 2012; 2014), is
likely a product of Cenozoic Alpine orogeny. Izquierdo-Llavall et al. (2019) confirmed that the
interpreted Alpine rotations correspond well with the amount of shortening reconstructed in
Meso-Cenozoic basins. The best preserved and most continuous outcrop in the Central
Iberian's outer arc is not a Variscan structure, casting doubt that Central Iberian curve's is





related to Variscan kinematics. The results are also a reminder that the regioanl effects of
Alpine deformation are often underestimated, especially close to the major Iberian Alpine fronts:
the Pyrenees, Iberian Ranges, and the Betics.

Overall, new paleomagnetic data from the Central Iberian curve and nearby areas reveal

pervasive late Carboniferous remagnetizations and regional vertical-axis rotations of the same
sense and magnitude to those expected for the southern arm of the Cantabrian Orocline. The
new paleomagnetic data indicate that a post ~320 Ma formation for the Central Iberia curve due
to vertical-axis rotations is not supported (Pastor-Galán et al., 2016). The distribution in space
and time of paleomagnetic results discards the formation of the Central Iberian curve as a
product of Variscan gravitational collapse (E1, ~330-317 Ma) or concomitant to the Cantabrian
Orocline (C3). So far, no pre-E2 paleomagnetic component has been found, and consequently,
paleomagnetic data cannot reject an early orogenic origin for the inner arc of the Central Iberian
curve (C1-C2, older than 330 Ma).

From a structural geology point of view, the Central Iberian curve does not display the

classic geometries and structural interference patterns as found in other established oroclines
(i.e., those systems that involve differential vertical-axis rotations, e.g. Li et al., 2012; van der
Boon et al., 2018; Meijers et al., 2017; Rezaeian et al., in press). The geometry and structural
behaviour of oroclines should resemble, at the crustal-scale, a regional vertical-axis fold
preserved in plan-view, either formed by buckling (e.g. Johnston et al., 2001) or bending (e.g.
Cifelli et al., 2008) mechanisms. In oroclines, pre-existing structures tend to follow fold trends
around the curvature (e.g. Rosenbaum, 2014; Li et al., 2018). In addition, orocline cores tend to
preserve radial structures and shortening patterns in the inner arc and orocline parallel shear
zones and extension structures in their outer arc (e.g. Ries and Shackleton, 1976; Eichelberger
and McQuarrie, 2015), similar to what is observed in multilayer folds (e.g. Fossen, 2016).

The structural geometry of the Central Iberian curve lacks such patterns.

Paleomagnetism from the Iberian Ranges indicate that the Cantabrian and West Asturian
Leonese zones do not follow the Central Iberian curve, instead they continue their NWW-SEE
trend into the Mediterranean in what it is now the Betic chain (Rodríguez-Cañero et al., 2018;
Jabaloy-Sánchez et al., 2018; van Hinsbergen et al., 2020). Structural trends in the Ossa
Morena and the South Portuguese Zone do not show any change in along-strike structural trend
that supports large-scale CW rotations (e.g. Pérez-Cáceres et al., 2015; Quesada et al., 2019),
whereas existing paleomagnetic data from those zones (Leite Mendes et al., in press) support a
model of CCW rotation associated with the broader southern arm of the Cantabrian Orocline. In
the Central Iberian and Galicia Tras-os-Montes zones, the trend of curvature is irregular (see C1





fold patterns in Fig. 2A) and nowhere are the expected inner and outer arc-related structures
preserved (e.g. Dias da Silva et al. in press).

The curved shape of C1 fold axes in the Central Iberian zone is better explained by fold

interference patterns than vertical-axis rotations (e.g. Pastor-Galán et al., 2019b). Moreover, the
curved shape of the Galicia Tras-os-Montes allochthonous nappe, which was emplaced orogen
parallel, shows no evidence of vertical-axis rotation related structures (Fig. 2A; Dias da Silva et
al., in press). Other authors describe the changes in trend around the Central Iberian curve
expressed by C1 folds (Fig. 2A) as the product of fold interference patterns (e.g. Gutiérrez-
Alonso, 2009; Palero-Fernández et al., 2015; Jacques et al., 2018b; Dias da Silva et al., in
press). Pastor-Galán et al. (2019b) showed that curved C1 folds in the Central Iberian Zone
around the Galicia Tras-os-Montes boundary (Fig. 2A) are coaxial with C3 folds after restoring
the effects of C2 and E1 deformation phases (Fig. 11A). Both C1 and C3 formed under similar
shortening directions. In the same area, Jacques et al. (2018b) found similar fold interference
patterns, in addition they described kinematic incompatibility with the expected CW rotations
that would have occurred if the Central Iberian curve was an orocline. In other areas of the
Central Iberian Zone, the curved shape of C1 folds has been described as an interference
between C1 structures and their reorientation caused by C3 shear zones (Fig. 2A; e.g. Palero-
Fernández et al., 2015; Dias et al., 2016), or alternatively the interference between C1, C3 and
the E2 structures (Fig. 2A; Gutiérrez-Alonso, 2009; Arango et al., 2013; Rubio Pascual et al.

2013).

Overall, new geometric and kinematic data favor the interpretation that the Central

Iberian curve is not a structure formed by differential vertical-axis rotation as was the Cantabrian
Orocline, but one formed as a consequence of several competing processes. It is clear from the
current data that a combination of several deformation events caused the orientation of
structures that today delineate the shape of the Central Iberia curve. These include: (1) the
northern part of the outer-arc is the product of an Alpine rigid block rotation instead of Variscan
differential vertical-axis rotation (Pastor-Galán et al., 2018); (2) the curvature of the Galicia Tras-
os-Montes allochthonous nappe reflects its original shape and could be defined as a primary
curve (see Weil and Sussman, 2004), since it was emplaced orogen parallel and shows no sign
of vertical-axis rotations at any time (fig. 2A; Dias da Silva et al., in press); (3) Structural
analysis shows that fold interference patterns explain the geometry of the curved trends of
Central Iberian Zone's C1 folds (Pastor-Galán et al., 2019b), whose kinematics are incompatible
with the required CW rotations expected if the curve is an orocline (Jacques et al, 2018b).




## 6 Get Back: Ideas flowing out and endless questions


The pioneering works in the last decade that resurrected the idea of a Central Iberian
curve, speculated that both the Cantabrian and Central Iberian zones buckled together as
secondary oroclines (Fig. 12; Martínez-Catalán 2011; Shaw et al., 2012, 2014; Shaw and
Johnston, 2016; Carreras and Druguet, 2014). Later, Martínez Catalan et al. (2014) and Díez
Fernández and Pereira (2017) reformulated Martínez-Catalán's 2011 hypothesis and proposed
that the Central Iberian curve formed as an orocline between 315 and 305 Ma, and assigning
the Cantabrian Orocline a time frame between 305 and 295 Ma (Fig. 12). The proposed tectonic
mechanisms to support these early kinematic models are varied: (1) buckling of a ribbon
'Armorican' continent (Fig. 12A; Shaw et al., 2014; 2016); (2) buckling of a completely formed
Variscan orogen during a putative 'Pangea B' to 'Pangea A' transition in the late Carboniferous
(Fig. 12B; Carreras and Druguet, 2014; Martínez-Catalán et al., 2011); (3) indentation of
Laurussia into Gondwana during the early stages of collision (at present day SW Iberia, South
Portuguese Zone), producing first the Central Iberian curve as a mega drag-fold during
Carboniferous times and then slightly later the Cantabrian Orocline as a consequence of an
indentation process (fig. 12C; Simancas et al., 2013).
The reviewed data in sections 4 and 5 contradict the aforementioned hypotheses.
Paleomagnetism and structural patterns (section 5; Fig. 6-11) disagree with the necessary CW
rotations required to support a late Carboniferous orocline origin for the Central Iberian curve
(Models in Fig. 12A and B). In addition, the sense and magnitude of the vertical-axis rotations
observed in SW Iberia (Fig. 10) imply that the South Portuguese (Avalonian segment) and Ossa
Morena zones moved together with the southern limb of the Cantabrian Orocline during the
Pennsylvanian and Early Permian. This means that the South Portuguese Zone was already
parallel to the general trend of the Variscan orogen prior to Cantabrian Orocline formation,
implying the lack of a Laurussian rigid indenter into Gondwana (e.g. Simancas et al., 2013). This
discrepancy leaves orogen-parallel terrane transport as a possible explanation to the kinematics
observed in Ossa Morena and South-Portuguese Zones (e.g. Pérez-Cáceres et al., 2016). At
the same time, paleomagnetism from SW Iberia backs the hypothesis of a Greater Cantabrian
Orocline extended into both Gondwana and Laurussia in its northern and southern limbs (Fig.
10; Pastor-Galán et al., 2015b).
In spite of the kinematic constraints and structural patterns, which do not support a
vertical-axis origin for the Central Iberian curve in Late Carboniferous time, other geometric
constraints remain challenging. The curved shape of the aeromagnetic and gravity anomalies of
Iberia are real (Fig. 5). These striking patterns could be due to Variscan-Alpine structural
interference, for example the previous example from the Iberian Ranges, but currently there is
not enough data to rigorously test this hypothesis. Shaw et al. (2012) supported their hypothesis
of a secondary orocline by assuming that paleocurrents were parallel through Iberia during
Ordovician times. However, some of the observed deflections in the paleocurrents studied by
Shaw et al. (2012; see Fig. 3) are also explained by Alpine vertical-axis rotations (the case of
the Iberian ranges) and fold interference patterns (SE of the Central Iberian Zone). Others
(Central and SW of the Central Iberian Zone) may be explained by a local response to basin
architecture (Fig. 3), where paleo-flow directions would trend toward the deepest basin
throughs. The Ordovician basin architecture of Iberia allows for opposite directed paleocurrents
from both sides of such throughs (Fig. 3). However, the Early Paleozoic basin architecture in
Iberia and their local deformation events require further research (Sánchez-García et al., 2019).
Although kinematic evidence is still scarce for the earliest Variscan movements, we
argue that pre-orogenic physiographic features, such as the opening of a marginal restricted
ocean between Gondwana and its distal platform at 395 Ma (Fig. 13A; Pin et al., 2002;
Gutiérrez-Alonso et al., 2008b; Arenas et al., 2016) explains the rounded shape of the Galicia
tras-os-Montes curve as a primary arc. During the collision, the latter irregularity would cause
the orogen-parallel emplacement of allochthonous nappe (Fig. 13B; Dias da Silva et al., in
press) and the left-lateral movements of the Ossa Morena and South Portuguese Zones in SW
Iberia (Fig 13A, B, C; Quesada, 2019). During the late Carboniferous, possibly due to a plate
reorganization during the final amalgamation of Pangea (Fig. 13D; e.g. Gutiérrez-Alonso et al.,
2008a; Pastor-Galán et al., 2015a), the far-field stress-field likely changed and buckled the
entire orogen around a vertical axis (Gutiérrez-Alonso et al., 2004), including both the
Gondwana and Laurussia margins (Fig. 13E; Pastor-Galán et al., 2015b).

## Acknowledgements

DPG thanks the extraordinary hard work, patience and endurance of the Utrecht
University students that embraced and enjoyed studying the kinematics of the central Iberian
curvature: Thomas Groenewegen, Bart Ursem, Daniel Brower, Mark Diederen and Bruno Leite-
Mendes. DPG acknowledges FRIS and CNEAS for the continous financial support. GGA is
supported by Spanish Ministry of Science, innovation and universities under the project
IBERCRUST (PGC2018-096534-B-I00) and Russian Federation Government grant no.
14.Y26.31.0012. This paper is a contribution to the IGCP no. 648 "Supercontinent Cycles and
Global Geodynamics". 50 years ago four fabulous guys let it be and never got back.



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

## 1464    Captions

Fig. 1 Simplified paleogeographic map of the Variscan-Alleghanian orogeny prior to the
Jurassic break-up of Pangea, with the major orogenic curves labeled. Note, this map represents





Iberian outcrops without taking account of the post Jurassic Alpine deformation (see text for
details e.g. Gong et al., 2008; Pastor-Galán et al., 2018). Slightly darker colors in the Variscan
belt indicate present-day European and African outcrops (modified after Martínez-Catalán et al.,
2009; Weil et al., 2013).

Fig. 2 A) Present-day configuration of the tectonostratigraphic zones of the Iberian

Variscides and its major structures. White areas represent Mesozoic and Cenozoic cover. B)
Three competing geometric proposals for the Central Iberian curve. 1) A disharmonic curvature,
up to 160˚ at the outer arc but much less pronounced at the inner arc (Aerden, 2004); 2) A
harmonic, but more open curvature as suggested by Martínez Catalán (2012); 3) an isoclinal
curvature model (Shaw et al., 2012). C) Distribution of the E1 (in migmatitic domes) and C3 to
post-C3 granitoids in the NW of Iberia (modified from Pastor-Galán et al., 2016)

Fig. 3 A) Stratigraphic synthesis of the Gondwanan platform series in NW Iberia.

Cantabrian Zone columns are after Aramburu et al., 2002; Bastida, 2004; Murphy et al 2008;
Pastor-Galán et al., 2013a; 2013b. Iberian Range follows Gozalo et al., 2008; Mergl and
Zamora, 2012 and Calvín and Casas, 2014. West Asturian Leonese Zone stratigraphy is after
Pérez-Estaún, 1990; Marcos, 2004; Martínez-Catalán et al., 2004a; Gutiérrez-Marco et al.,
2019. Central Iberian Zone follows Díez-Balda,1986; Valladares et al., 2000; Díez Montes,
2007; Martínez-Catalán et al., 2004b; del Moral and Sarmiento, 2008; García-Arias et al., 2018.
B) Ordovician paleocurrents orientations, modified from Shaw et al., 2012. C) Schematic basin
architecture inferred from the stratigraphic compilation.

Fig. 4 A) The kinematic evolution of the Cantabrian Orocline in its core, the Cantabrian

Zone, inferred from total least squares (TLS) orocline tests (Pastor-Galán et al. 2017). B) Shows
three orocline (strike) tests used to constrain the kinematics of the Cantabrian Orocline. The
Ordovician paleocurrents, which predate any orogenic movement, recorded the complete
vertical-axis rotation history and yields a slope (m) of ~1. The Moscovian paleomagnetic data
(from Weil et al., 2013; Pmag.), which postdates the main orogenic phases (C1, C2 and E1) and
is coeval with C3, shows a slope of ~1. The Gzhelian joint sets (from Pastor-Galán et al., 2011)
orocline test shows a slope of ~0.5, which indicates that half of the orocline was already formed
at ~304 Ma.

Fig. 5 A) Aeromagnetic map of Spain (Ardizone et al., 1989, for Spain and the World

Digital Magnetic Anomaly Map (WDMAM project) and Portugal (modified from Martínez Catalán,
2012 and Martínez Catalán et al., 2015), showing the possible trace of the Central Iberian
curve. B) Bouguer anomalies of the Iberian Peninsula, modified from Ayala et al., 2016. Gravity
anomalies do not reflect the geometry of the Cantabrian Orocline nor the Central Iberian curve





but are related to the Cenozoic Alpine lithospheric structure.
Fig. 6 Paleomagnetic studies related to the Cantabrian Orocline and the Central Iberian
curve: (1) Synthesis of paleomagnetism in the core of the Cantabrian Orocline (see Weil et al.,
2013); (2) Permian (eP) components synthesized in Weil et al. (2010); (3) Ordovician volcanics
and limestones (Laquiana) in the boundary between the West Asturian-Leonese and Central
Iberian Zones (Fernández-Lozano et al., 2016); (4) Devonian sedimentary sequences and
Permian subvolcanics in the Iberian ranges (Pastor-Galán et al., 2018); (5) Permian dykes and
sills (Calvín et al, 2014); (5) Anatectic granites (E1) and mantle derived granitoids (C3) from
Tormes Dome and Central System (Pastor-Galán et al., 2016); (6) Cambrian limestones from
Tamames (N) and los Navalucillos (S) (Pastor-Galán et al., 2015a); (7) Ediacaran-Early
Cambrian sedimentary rocks in the southern sector of the Central Iberian Zone (Pastor-Galán et
al., 2017b); (8) Almadén volcanics from the Central Iberian Zone (Perroud et al., 1991; Parés
and van der Voo, 1992; Leite Mendes et al. in press) and Volcanic rocks from southern Ossa
Morena and the South Portuguese Zone (Leite Mendes et al, in press)
Fig. 7 Magnetization components with a positive reversal test in the extensional
anatectic granites of Tormes (A) and Martinamor Domes (B). This component is interpreted as
primary with a magnetization age of >318 Ma (Pastor-Galán et al., 2016). C) Distribution of
directions and VGPs and statistical parameters from both domes combined.
Fig. 8 Cartoon depicting the different vertical axis rotation events that occurred in the
Cantabrian Zone and the Iberian Range, modified from Pastor-Galán et al. (2018). (A) Original
quasilinear Variscan Orogenic belt, B) Formation of the Cantabrian Orocline around the
Carboniferous–Permian boundary after a ~70° counterclockwise rotation in the Southern branch
of the Cantabrian Zone and the Iberian Range. This rotation matches the rotation for the
Cantabrian Orocline, see the fit of the Iberian Range Component #2 in the orocline test for the
Cantabrian Zone (below). C) Post Permian (Cenozoic) rotation of ~22° clockwise (CW) likely
produced by differential shortening during the Alpine orogeny (Izquierdo-Llavall et al., 2018).
Below, the global magnetic polarity time scale for the Pennsylvanian and Cisuralian (following
Ogg et al., 2016). TLS = Total Least Squares. Note that once the 22° CW rotation in the Iberian
Range is corrected, components #2, #1, and P fit as expected with the APWP for the southern
limb of the orocline (Pastor-Galán et al., 2016).
Fig. 9 Compilation of the directional distributions and average declinations with
parachute of confidence (Δ Declination) in sites around the Central Iberian curve (see Fig. 6).
The results show general CCW rotations in contrast to the expected CW if the Central Iberia
curve formed by vertical-axis rotations (see text). Results are compared with the expected





declinations if those sites were part of the Cantabrian Orocline following the methodology
described in Pastor-Galán et al., 2017b.
Fig. 10 Orocline test of the Cantabrian Orocline (Weil et al., 2013) compared with the
magnetizations found in the adjacent Laurussian segments of the orogen: Ireland (Pastor-Galán
et al., 2015b) and the South Portuguese Zone (Leite Mendes et al., in press)
Fig. 11 Structural analysis of mullions in the Central Iberian Zone (after Pastor-Galán et
al., 2019b) A) Photograph (with card scale, 10cm) of a bedding plane surface showing the
mullions and photograph analysis. B) Interpretation of the outcrop with fold axis traces depicting
the deformation phase responsible for each structure. C) Result of retro-deformation of mullions
in the Mogadouro road section modified from Pastor-Galán et al. (2019b). Unfolding the effects
of D3 on D2 mullions. Unfolding the effects of C3 and E1 in C1 mullions.
Fig. 12 Pionering hypothesis for the Central Iberian curve. Note that none of them fulfill
the most recent geometric and kinematic criteria. A) Simplified ribbon continent model after
Johnston et al. (2013) and Shaw and Johnston (2016). B) Dextral mega-shear model from
Martínez-Catalán (2011). C) Kinematic model with indentation and left-lateral shearing after
Simancas et al. (2013)
Fig. 13 Preliminary kinematic proposal for the Iberian Variscides. A) Pre-colisional stage
after the opening of the Galicia Tras-os-Montes restricted seaway (e.g. Pin et al., 2002;
Gutiérrez-Alonso et al., 2008a; Arenas et al., 2016). The irregular shape of the margin and the
younging westwards deformation front (e.g. Daleyer et al., 1997) resulted in tectonic escape
towards the still open Rheic Ocean (e.g. Braid et al., 2011; Murphy et al., 2016). B) After closure
of the Rheic ocean, C1 and C2 structures formed. The Galicia Tras-os-Montes was emplaced
orogen parallel (e.g. Martínez-Catalán et al., 1990; Dias da Silva et al., in press), preserving the
shape of the seaway, i.e. a primary arc. C) The gravitational collapse of the orogen produced
widespread anatexis and folding interference in the hinterland and the emplacement of the
foreland fold-and-thrust belt. D) At Pennsylvanian times a change in the far-field stress buckled
the Variscan belt around a vertical axis (see Gutiérrez-Alonso et al., 2008; Weil et al., 2013;
Pastor-Galán et al., 2015a for details), creating new interference patterns and a lithospheric
scale response (see Gutiérrez-Alonso et al., 2004, 2011a; Pastor-Galán et al., 2012a). E) When
the orocline became too tight to keep rotating, new cross-cutting brittle structures (C4) formed
and minor extensional collapse (E2) occurred (e.g. Fernández-Lozano et al., 2019; Dias da
Silva et al., in press).



Fig. 1 Simplified paleogeographic map of the Variscan-Alleghanian orogeny prior to the
Jurassic break-up of Pangea, with the major orogenic curves labeled. Note, this map represents
Iberian outcrops without taking account of the post Jurassic Alpine deformation (see text for
details e.g. Gong et al., 2008; Pastor-Galán et al., 2018). Slightly darker colors in the Variscan
belt indicate present-day European and African outcrops (modified after Martínez-Catalán et al.,
2009; Weil et al., 2013).

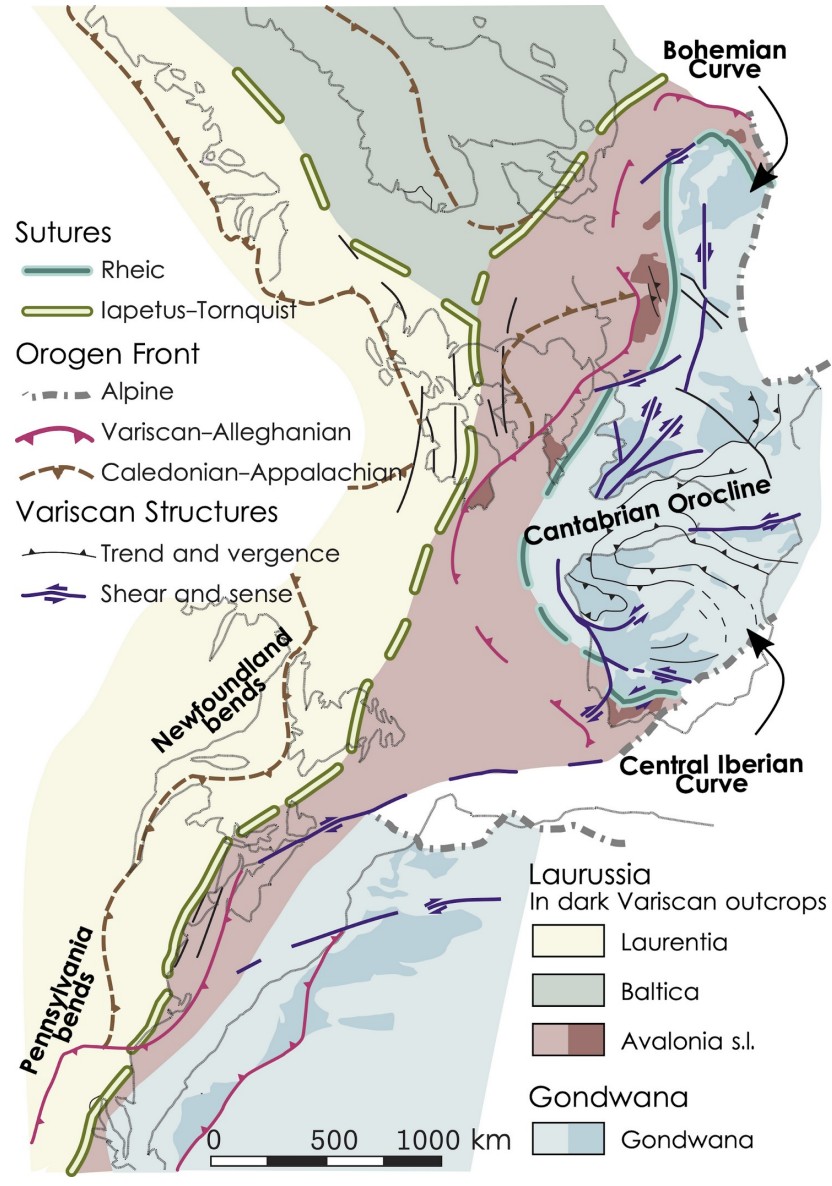


Fig. 2 A) Present-day configuration of the tectonostratigraphic zones of the Iberian
Variscides and its major structures. White areas represent Mesozoic and Cenozoic cover. B)
Three competing geometric proposals for the Central Iberian curve. 1) A disharmonic curvature,
up to 160˚ at the outer arc but much less pronounced at the inner arc (Aerden, 2004); 2) A
harmonic, but more open curvature as suggested by Martínez Catalán (2012); 3) an isoclinal
curvature model (Shaw et al., 2012). C) Distribution of the E1 (in migmatitic domes) and C3 to
post-C3 granitoids in the NW of Iberia (modified from Pastor-Galán et al., 2016)

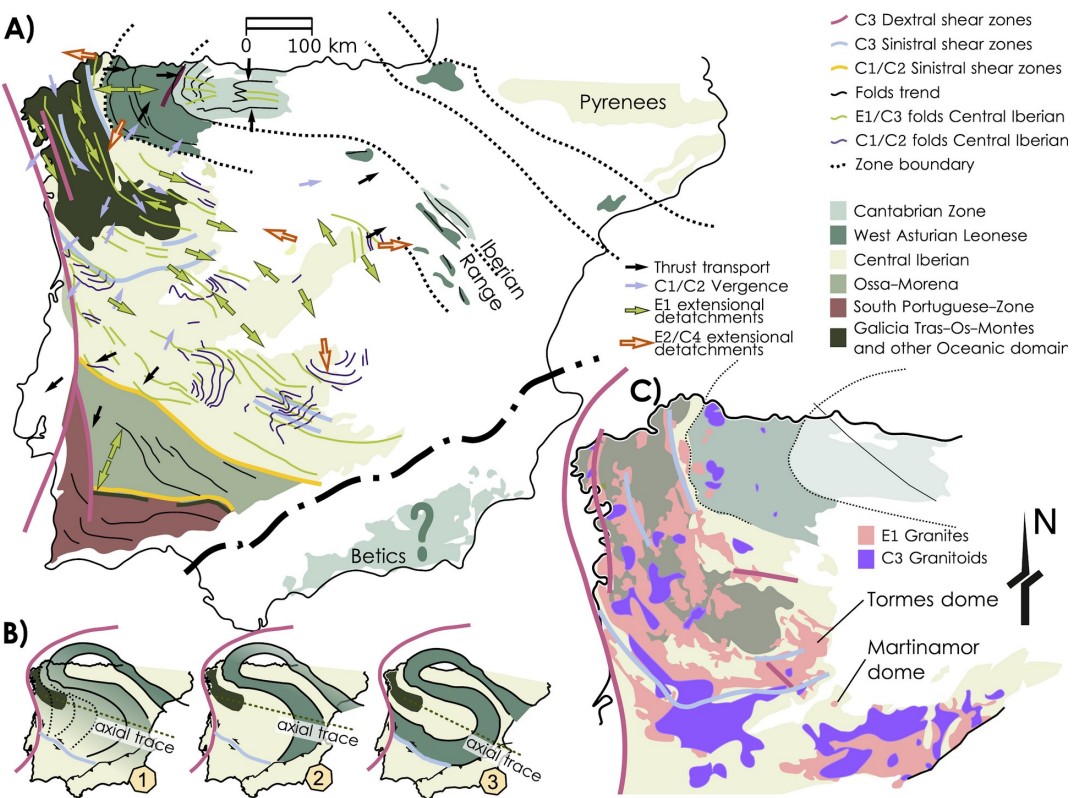




Fig. 3 A) Stratigraphic synthesis of the Gondwanan platform series in NW Iberia.
Cantabrian Zone columns are after Aramburu et al., 2002; Bastida, 2004; Murphy et al 2008;
Pastor-Galán et al., 2013a; 2013b. Iberian Range follows Gozalo et al., 2008; Mergl and
Zamora, 2012 and Calvín and Casas, 2014. West Asturian Leonese Zone stratigraphy is after
Pérez-Estaún, 1990; Marcos, 2004; Martínez-Catalán et al., 2004a; Gutiérrez-Marco et al.,
2019. Central Iberian Zone follows Díez-Balda,1986; Valladares et al., 2000; Díez Montes,
2007; Martínez-Catalán et al., 2004b; del Moral and Sarmiento, 2008; García-Arias et al., 2018.
B) Ordovician paleocurrents orientations, modified from Shaw et al., 2012. C) Schematic basin
architecture inferred from the stratigraphic compilation.

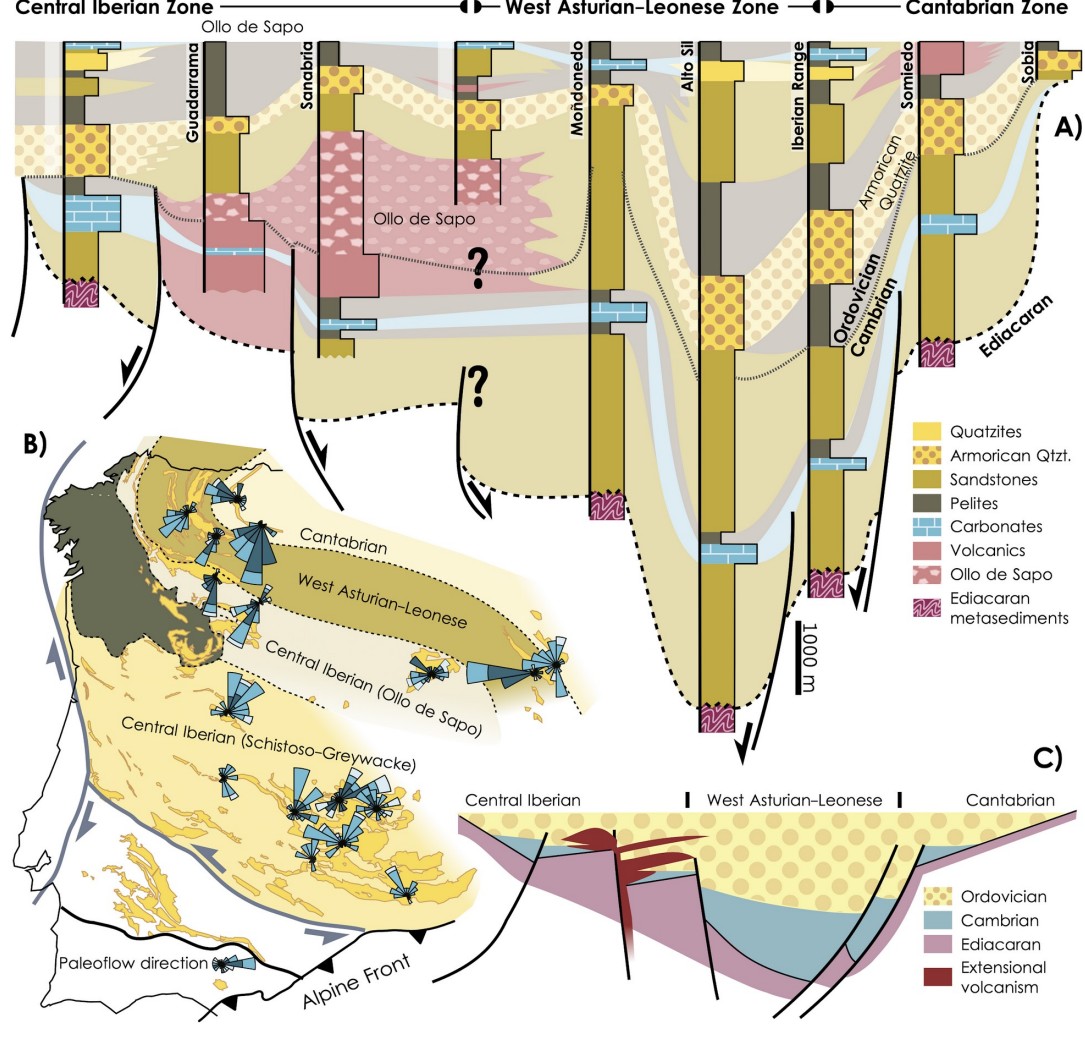




Fig. 4 A) The kinematic evolution of the Cantabrian Orocline in its core, the Cantabrian
Zone, inferred from total least squares (TLS) orocline tests (Pastor-Galán et al. 2017). B) Shows
three orocline (strike) tests used to constrain the kinematics of the Cantabrian Orocline. The
Ordovician paleocurrents, which predate any orogenic movement, recorded the complete
vertical-axis rotation history and yields a slope (m) of ~1. The Moscovian paleomagnetic data
(from Weil et al., 2013; Pmag.), which postdates the main orogenic phases (C1, C2 and E1) and
is coeval with C3, shows a slope of ~1. The Gzhelian joint sets (from Pastor-Galán et al., 2011)
orocline test shows a slope of ~0.5, which indicates that half of the orocline was already formed
at ~304 Ma.

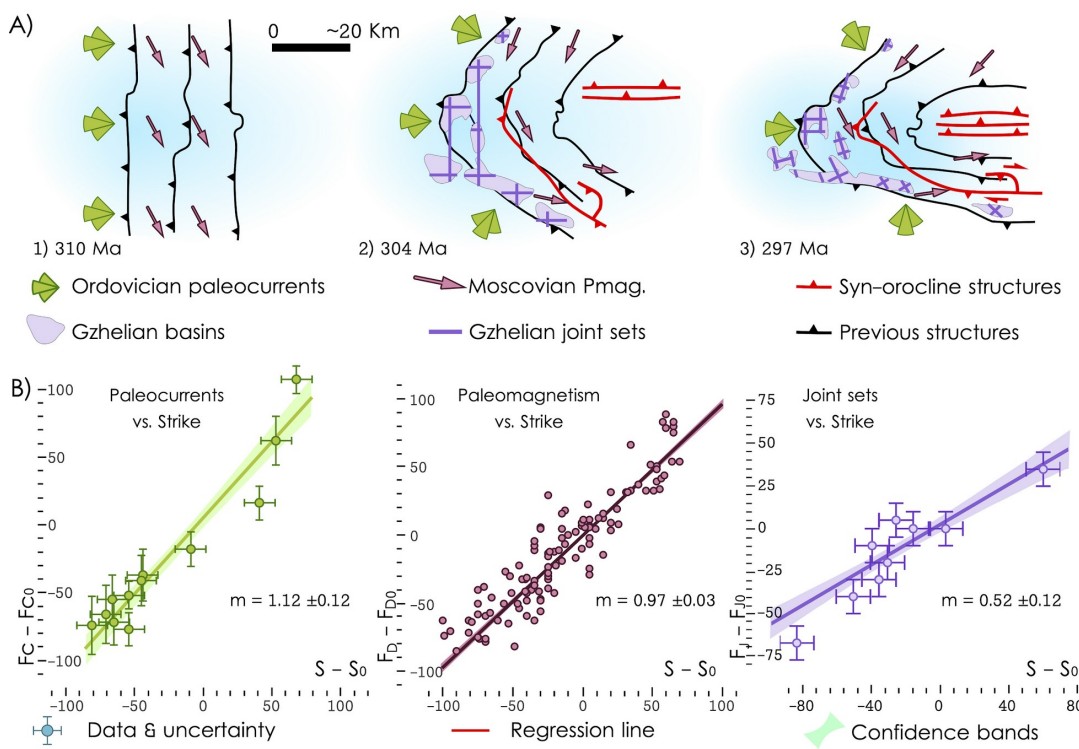


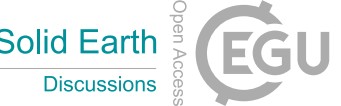

Fig. 5 A) Aeromagnetic map of Spain (Ardizone et al., 1989, for Spain and the World
Digital Magnetic Anomaly Map (WDMAM project) and Portugal (modified from Martínez Catalán,
2012 and Martínez Catalán et al., 2015), showing the possible trace of the Central Iberian
curve. B) Bouguer anomalies of the Iberian Peninsula, modified from Ayala et al., 2016. Gravity
anomalies do not reflect the geometry of the Cantabrian Orocline nor the Central Iberian curve
but are related to the Cenozoic Alpine lithospheric structure.

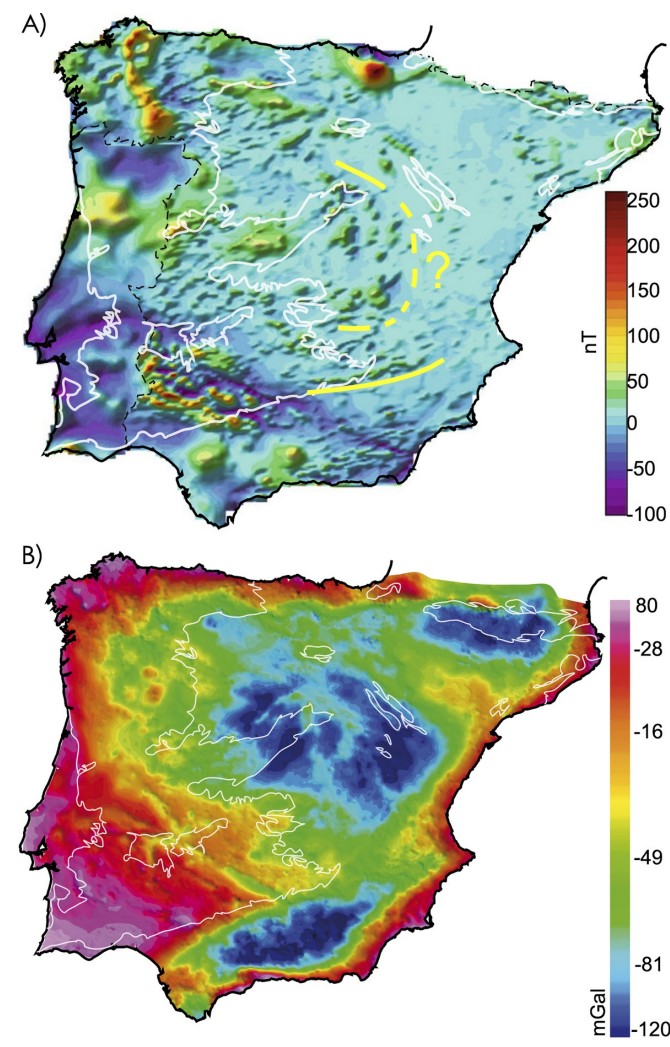

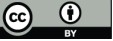


Fig. 6 Paleomagnetic studies related to the Cantabrian Orocline and the Central Iberian
curve: (1) Synthesis of paleomagnetism in the core of the Cantabrian Orocline (see Weil et al.,
2013); (2) Permian (eP) components synthesized in Weil et al. (2010); (3) Ordovician volcanics
and limestones (Laquiana) in the boundary between the West Asturian-Leonese and Central
Iberian Zones (Fernández-Lozano et al., 2016); (4) Devonian sedimentary sequences and
Permian subvolcanics in the Iberian ranges (Pastor-Galán et al., 2018); (5) Permian dykes and
sills (Calvín et al, 2014); (5) Anatectic granites (E1) and mantle derived granitoids (C3) from
Tormes Dome and Central System (Pastor-Galán et al., 2016); (6) Cambrian limestones from
Tamames (N) and los Navalucillos (S) (Pastor-Galán et al., 2015a); (7) Ediacaran-Early
Cambrian sedimentary rocks in the southern sector of the Central Iberian Zone (Pastor-Galán et
al., 2017b); (8) Almadén volcanics from the Central Iberian Zone (Perroud et al., 1991; Parés
and van der Voo, 1992; Leite Mendes et al. in press) and Volcanic rocks from southern Ossa
Morena and the South Portuguese Zone (Leite Mendes et al, in press)

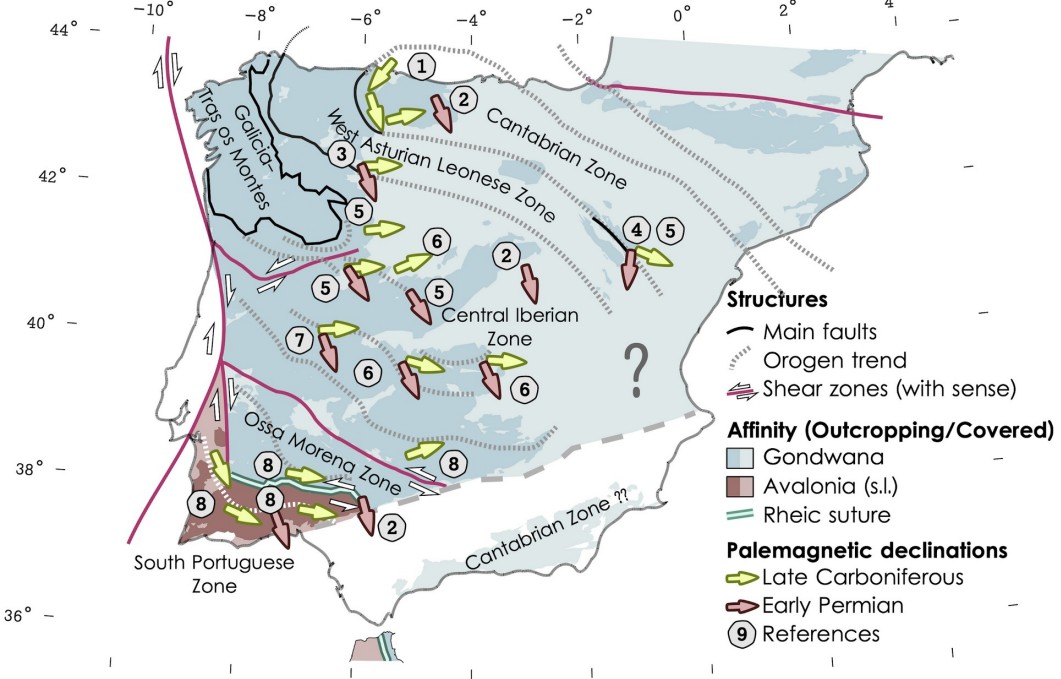




Fig. 7 Magnetization components with a positive reversal test in the extensional anatectic
granites of Tormes (A) and Martinamor Domes (B). This component is interpreted as primary
with a magnetization age of >318 Ma (Pastor-Galán et al., 2016). C) Distribution of directions
and VGPs and statistical parameters from both domes combined.

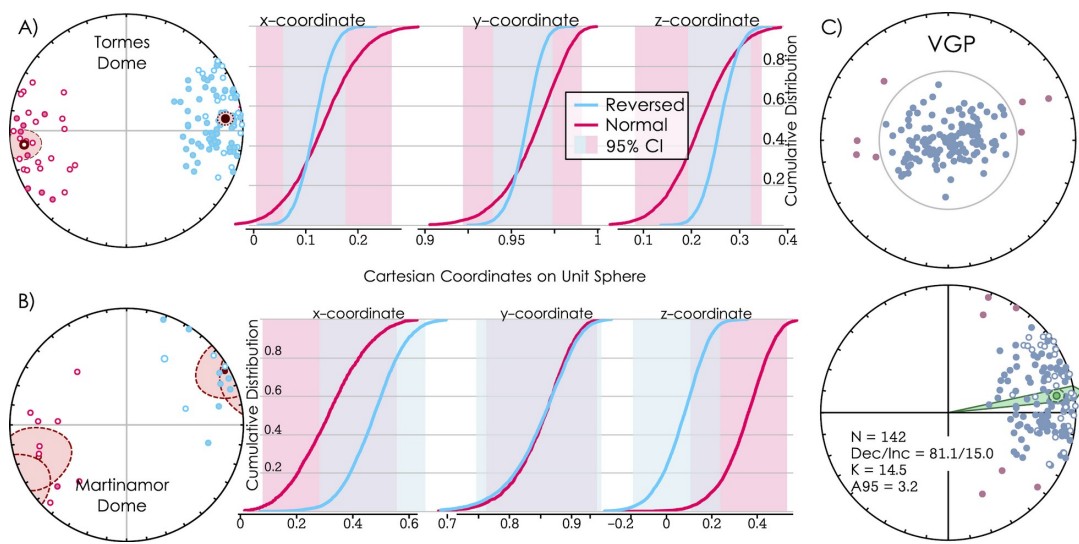

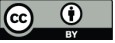



Fig. 8 Cartoon depicting the different vertical axis rotation events that occurred in the
Cantabrian Zone and the Iberian Range, modified from Pastor-Galán et al. (2018). (A) Original
quasilinear Variscan Orogenic belt, B) Formation of the Cantabrian Orocline around the
Carboniferous–Permian boundary after a ~70° counterclockwise rotation in the Southern branch
of the Cantabrian Zone and the Iberian Range. This rotation matches the rotation for the
Cantabrian Orocline, see the fit of the Iberian Range Component #2 in the orocline test for the
Cantabrian Zone (below). C) Post Permian (Cenozoic) rotation of ~22° clockwise (CW) likely
produced by differential shortening during the Alpine orogeny (Izquierdo-Llavall et al., 2018).
Below, the global magnetic polarity time scale for the Pennsylvanian and Cisuralian (following
Ogg et al., 2016). TLS = Total Least Squares. Note that once the 22° CW rotation in the Iberian
Range is corrected, components #2, #1, and P fit as expected with the APWP for the southern
limb of the orocline (Pastor-Galán et al., 2016)

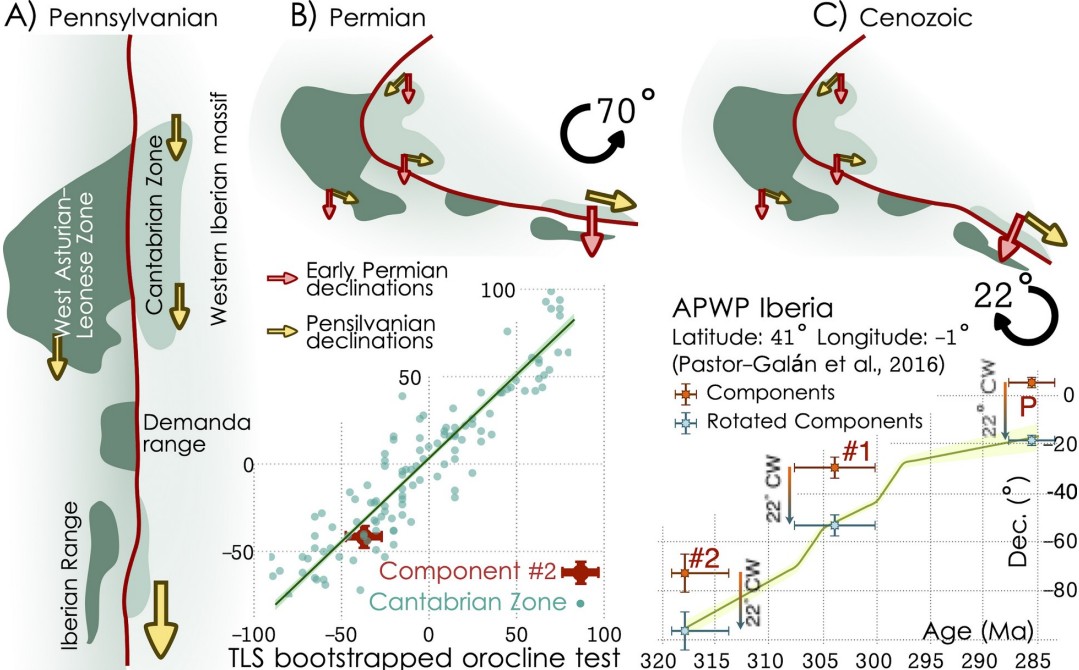





Fig. 9 Compilation of the directional distributions and average declinations with parachute of
confidence (Δ Declination) in sites around the Central Iberian curve (see Fig. 6). The results
show general CCW rotations in contrast to the expected CW if the Central Iberia curve formed
by vertical-axis rotations (see text). Results are compared with the expected declinations if
those sites were part of the Cantabrian Orocline following the methodology described in Pastor-
Galán et al., 2017b

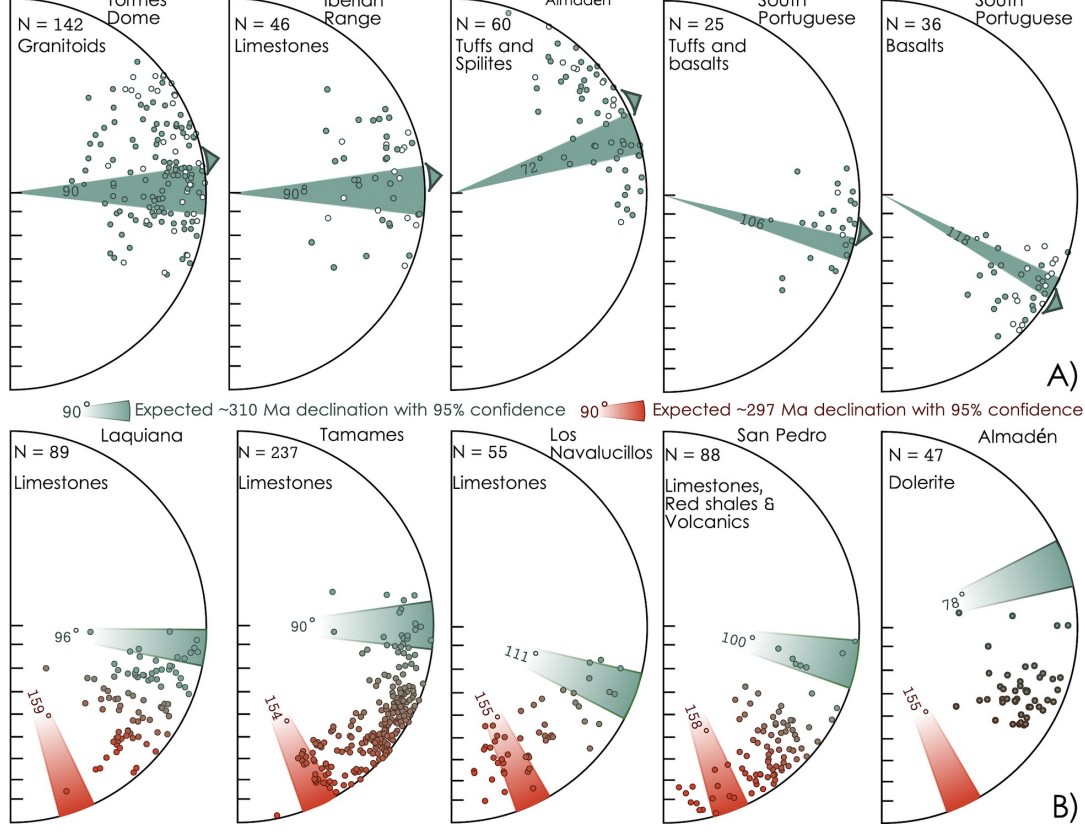




Fig. 10 Orocline test of the Cantabrian Orocline (Weil et al., 2013) compared with the
magnetizations found in the adjacent Laurussian segments of the orogen: Ireland (Pastor-Galán
et al., 2015b) and the South Portuguese Zone (Leite Mendes et al., in press)

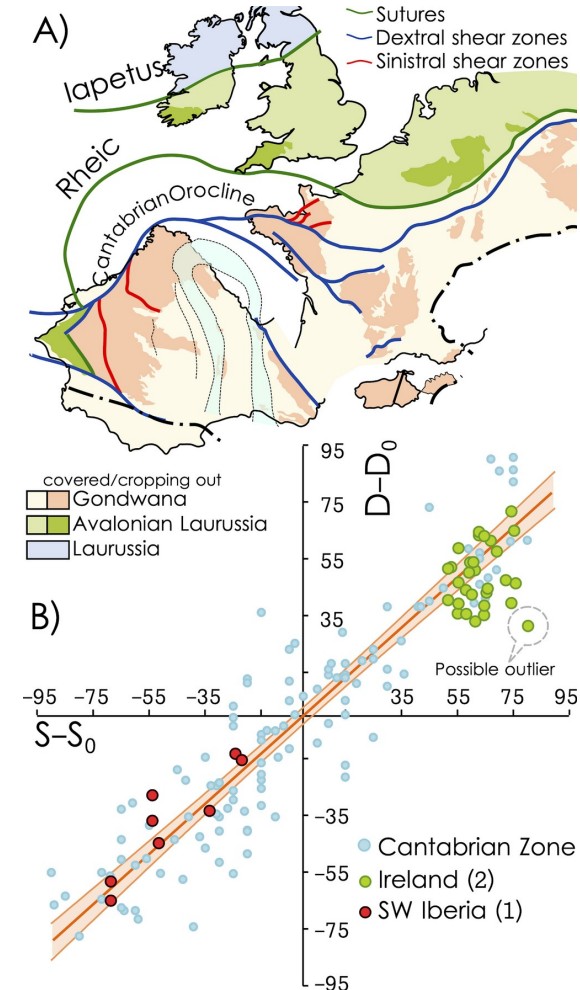




Fig. 11 Structural analysis of mullions in the Central Iberian Zone (after Pastor-Galán et
al., 2019b) A) Photograph (with card scale, 10cm) of a bedding plane surface showing the
mullions and photograph analysis. B) Interpretation of the outcrop with fold axis traces depicting
the deformation phase responsible for each structure. C) Result of retro-deformation of mullions
in the Mogadouro road section modified from Pastor-Galán et al. (2019b). Unfolding the effects
of D3 on D2 mullions. Unfolding the effects of C3 and E1 in C1 mullions.

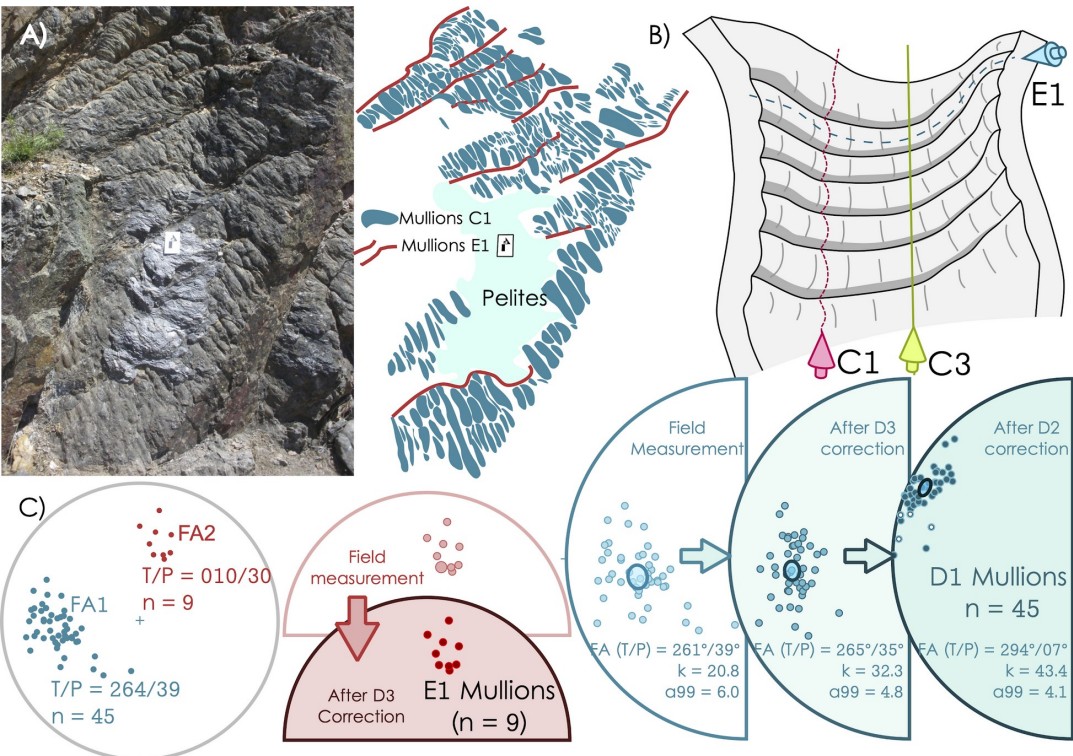




Fig. 12 Pioneering hypothesis for the Central Iberian curve. Note that none of them fulfill
the most recent geometric and kinematic criteria. A) Simplified ribbon continent model after
Johnston et al. (2013) and Shaw and Johnston (2016). B) Dextral mega-shear model from
Martínez-Catalán (2011). C) Kinematic model with indentation and left-lateral shearing after
Simancas et al. (2013)

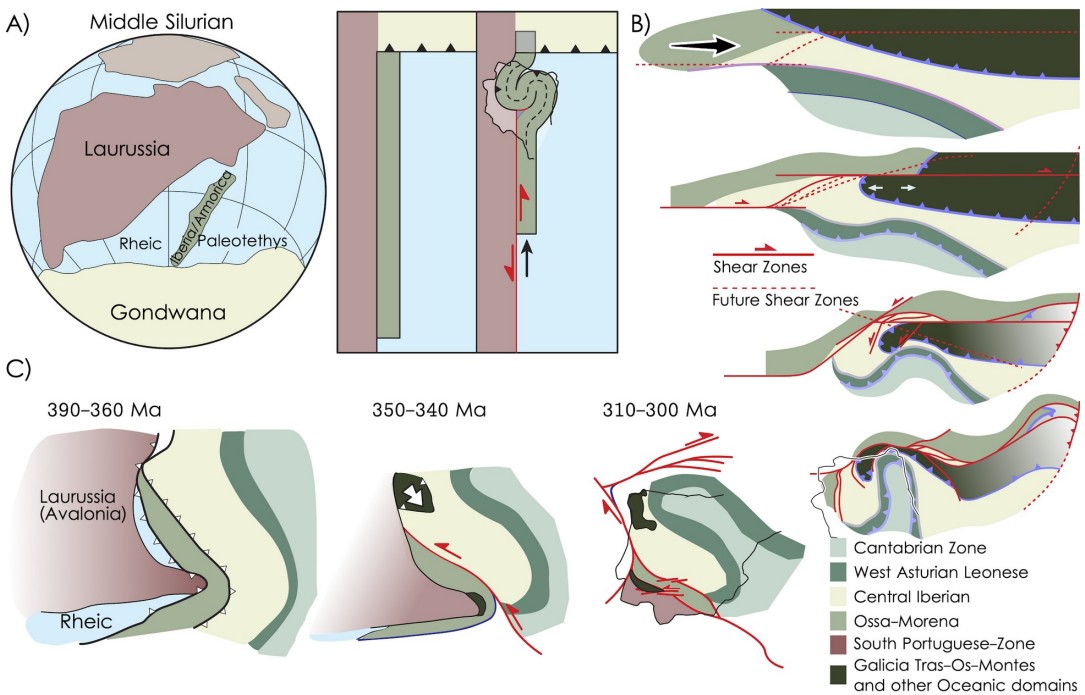



Fig. 13 Preliminary kinematic proposal for the Iberian Variscides. A) Pre-colisional stage
after the opening of the Galicia Tras-os-Montes restricted seaway (e.g. Pin et al., 2002;
Gutiérrez-Alonso et al., 2008a; Arenas et al., 2016). The irregular shape of the margin and the
youging westwards deformation front (e.g. Daleyer et al., 1997) resulted in tectonic escape
towards the still open Rheic Ocean (e.g. Braid et al., 2011; Murphy et al., 2016). B) After closure
of the Rheic ocean, C1 and C2 structures formed. The Galicia Tras-os-Montes was emplaced
orogen parallel (e.g. Martínez-Catalán et al., 1990; Dias da Silva et al., in press), preserving the
shape of the seaway, i.e. a primary arc. C) The gravitational collapse of the orogen produced
widespread anatexis and folding interference in the hinterland and the emplacement of the
foreland fold-and-thrust belt. D) At Pennsylvanian times a change in the far-field stress buckled
the Variscan belt around a vertical axis (see Gutiérrez-Alonso et al., 2008; Weil et al., 2013;
Pastor-Galán et al., 2015a for details), creating new interference patterns and a lithospheric
scale response (see Gutiérrez-Alonso et al., 2004, 2011a; Pastor-Galán et al., 2012a). E) When
the orocline became too tight to keep rotating, new cross-cutting brittle structures (C4) formed
and minor extensional collapse (E2) occurred (e.g. Fernández-Lozano et al., 2019; Dias da
Silva et al., in press).



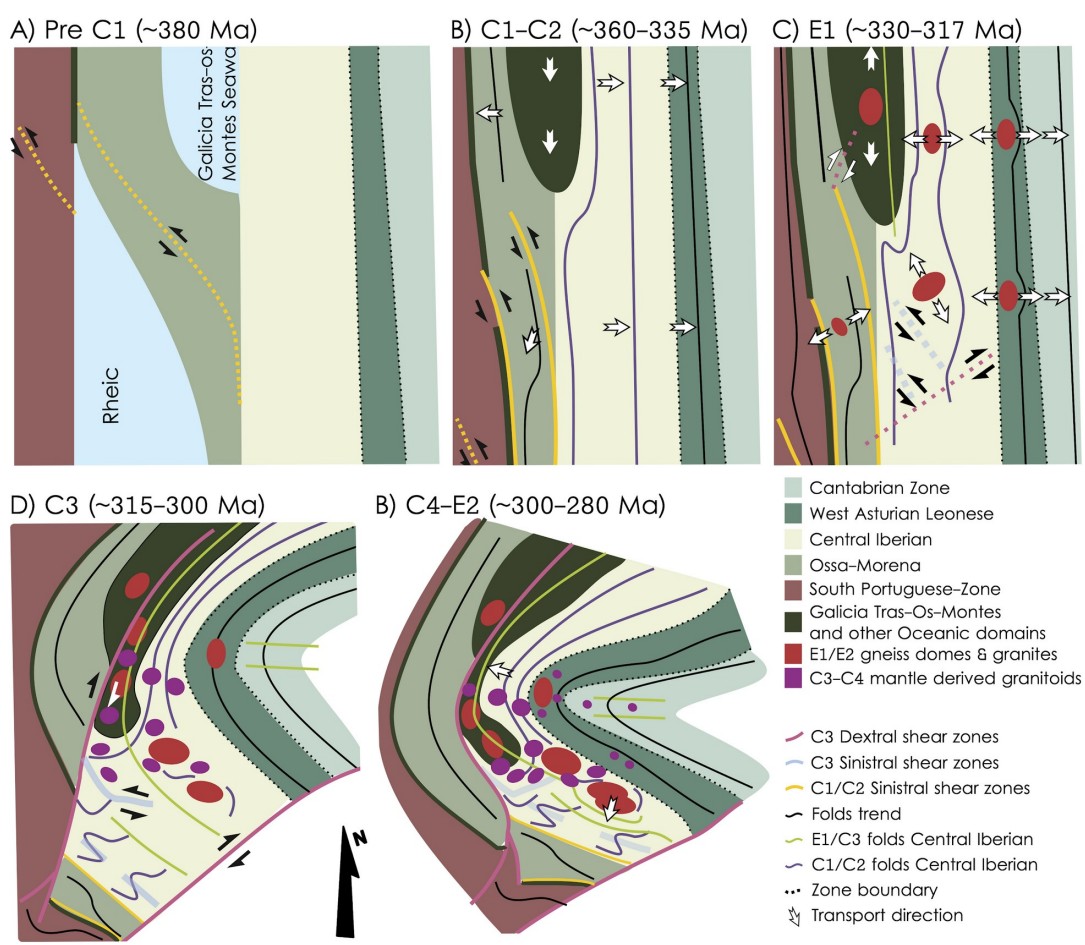