# Peer review of "The enigmatic curvature of Central Iberia and its puzzling kinematics"

_Solid Earth, 2020_

## Referee Comment (RC1) · Anonymous Referee #1 · 8 May 2020

The manuscript presents the main geological features of the Variscan belt in the Iberian Peninsula, in order to discuss the characteristics and origin of the double-arc geometry it presents. Based mainly on stratigraphic, structural and paleomagnetic data, the authors conclude that, unlike the Cantabrian Orocline, the Central Iberian curve is strictu senso not an orocline, but it shows rather a primary geometry at its core, while the characteristics of its external zone are due to superposed folding and the effects of the Alpine tectonics.

In my opinion, the work as a whole is excellent, and I recommend its publication in Solid Earth with minor modifications. In this case, the unorthodox format of the manuscript should not be an obstacle. It is not strictly a review paper, nor does it present original data. Instead, it makes a synthetic tour (without pretending to be exhaustive from the

bibliographic point of view) through the main characteristics of the Variscan belt in the center and western realms of the Iberian Peninsula. It also takes the essential data from the most recent relevant bibliography in order to show the non-oroclinal nature of the Central Iberian curve. From my point of view, the presented (and the available) structural information is not so relevant as to rule out that option (perhaps this should be better highlighted in the manuscript, as it is done with the paleocurrent data), but the paleomagnetic evidence is undoubtedly very robust. The interpretation presented by the authors in figure 13 is the most consistent with the available paleomagnetic information. For all those reasons, the article deserves to be published in this issue of Solid Earth.

I recommend some minor changes, mostly typing mistakes, grammar error in other cases. I indicate them in the annotated version of the manuscript that accompanies this report. There may be other errors, so I ask the authors to review the manuscript very carefully to remove them. Concerning the contents, I have included some comments in the annotated manuscript. I would like to stress here the importance of making a somewhat more important change. It refers to figure 11 (structural analysis of mullions ...). This figure is puzzling for several reasons. To begin with, the scale difference between figure 11 (study of a small outcrop) in relation to the large scale of all the other figures is very striking. Note that there also a great difference between figure 11 and the large scale on which the interpretation of the tectonic evolution of the studied region is focused. But, even more important is the fact that the data shown in that figure 11 are not original from this work. In addition, the information of a single outcrop is presented, instead that of a set of stations strategically located throughout the arch (as is done in the manuscript with the paleomagnetic or stratigraphic data). Furthermore, the structural argument that the figure tries to illustrate is presented quite clearly in the main text, together with the necessary reference to the original source of the data. Therefore, I strongly recommend deleting figure 11. It would be possible to make that change without any modification in the text (which proves that it is an avoidable figure), implying only a reorganization in the numbering of the following two
figures.

Please also note the supplement to this comment:
https://www.solid-earth-discuss.net/se-2020-51/se-2020-51-RC1-supplement.pdf

---

## Referee Comment (RC2) · Dominique Jacques (Referee) · 13 May 2020

The manuscript presents an excellent review of a long-standing debate on the nature and origin of the conspicuous Variscan orocline(s) in the Iberian Massif. Firstly, the evolution of the debate on the nature of the Central Iberian curve from its onset towards its current state is accurately presented. Secondly, the relevant paleomagnetic, structural, stratigraphic and sedimentological data that have been collected are discussed. And finally, the authors critically evaluate the different hypotheses of the Central Iberian curve considering the existing kinematic and paleomagnetic criteria. Based on all the literature and performed analyses they attain the most plausible conclusion: the Central Iberian curve should not be considered as a veritable orocline. At its core the Central Iberian curve is a primary feature of the Variscan orogenic belt that does not

show any late Carboniferous to early Permian vertical-axis rotation, while in its external hinge zone it is an 'artefact' from recent Alpine tectonics.

The paper is well-written, clearly structured, its bibliography is extensive and up-to-date and the figures are relevant and of high scientific quality. Hence, I can strongly recommend the manuscript for publication in Solid Earth, taking into account some minor revisions.

Besides mostly spelling mistakes (see annotated pdf), the most significant of these revisions concerns the supposed existence and nature of the post-Variscan C4 stage. The authors suggest a subdivision between a C3 (315-300Ma) and C4 (300-280Ma) deformation stage, although both stages are characterized by N-S shortening and the widespread development of shear zones. The main distinction is the contrast of mainly ductile C3 folding and C4 brittle faulting. I would argue that a gradual evolution from ductile to brittle deformation as the orogeny is uplifted and cools does not warrant a separate deformation stage if the dynamics remain similar. Figure 13E even suggests continued tightening of the orocline, prior to E2. In addition, I also disagree with the notion that the early Permian (i.e. posterior to oroclinal buckling) was characterized by N-S compression. Instead, literature generally agrees that Gondwana-Laurussia convergence during this time period was NW-SE to W-E directed. In the southern Variscan orogeny this led to NW-SE transpression and strike-slip deformation, especially observed in Stephanian-Permian intramontane basins. Simultaneously, to the west of the Variscan orogeny, frontal collision of Gondwana and Laurussia led to the formation of the Alleghenides, Mauritanides and the Ouachita-Marathon-Sonora orogenic belts. The authors do not take into account the related literature, which are not in accordance with their geodynamic model in Fig. 13E. Hence, I would suggest that the authors either (i) limit themselves to the geodynamics of oroclinal buckling (C3 – 315-295Ma) in their interpretation, or (ii) strongly extend their literature study on the kinematics of post-Variscan deformation (295-280Ma). A more detailed account of the above comments and related literature have been added to the annotated pdf in

attachment.

Please also note the supplement to this comment:
https://www.solid-earth-discuss.net/se-2020-51/se-2020-51-RC2-supplement.pdf
—————————————————————

---

## Author Comment (AC1) · 18 May 2020

Response to Anonymous Referee #1 (R1 hereafter)

We want to thank R1 the time invested in the review and the nice words about our manuscript.

We have followed all the suggestions and corrected all typos marked in the annotated PDF. We have also reviewed the text as indicated by R1 and identified some other mistakes.

R1 points out that the available structural information is not so relevant as the paleo-magnetic one to rule out the Central Iberian curve as a secondary orocline. Following R1 recommendation, we highlighted the milestones and limitations of the structural

analyses in the Central Iberian curve in section 6 Get Back: Ideas flowing out and endless questions:

'In addition are the curved traces of C1 fold-axes, whose geometry and kinematics are reasonably well constrained around Galicia Tras-os-Montes (e.g. Dias da Silva et al., in press), but in many other areas their strong curvature remain largely unstudied (Fig. 2) and therefore, to date we can only speculate on their origin."

Finally, R1 is right on the previous figure 11. We agree it is not a fully necessary figure and their content is properly described and referred in the text. We have removed this figure from the revised version of the paper.

Daniel Pastor-Galán on behalf of all co-authors

---

## Author Comment (AC2) · 18 May 2020

Response to Referee #2 (Dominique Jacques)

We want to thank Dominique Jacques for the thorough review and insights provided.

We have followed all the suggestions and corrected all typos, spelling mistakes and suggestions annotated in the attached PDF.

Apart from stylistic and orthographic corrections Dominique Jacques has some in detail comments:

**DJ: "the most significant of these revisions concerns the supposed existence and nature of the post-Variscan C4 stage. The authors suggest a subdivision between

a C3 (315-300Ma) and C4 (300-280Ma deformation stage, although both stages are characterized by N-S shortening and the widespread development of shear zones. The main distinction is the contrast of mainly ductile C3 folding and C4 brittle faulting. I would argue that a gradual evolution from ductile to brittle deformation as the orogeny is uplifted and cools does not warrant a separate deformation stage if the dynamics remain similar. Figure 13E even suggests continued tightening of the orocline, prior to E2."**

Here Dominique is right, C3 and C4 are roughly N-S shortening events (in present day coordinates) , that likely respond to the same strain field. The same could be said in the Iberian sector of the Variscan belt about C1 and C2. This paper is a review and we followed the most recent terminology for the Central Iberian Zone for consistency (Martínez Catalán et al., 2014). In addition, we think that phase naming does not need to reflect only dynamic or kinematic separate events, but also structural styles or meta-morphic conditions changing progressively in accordance with the orogenic evolution. In this sense, we think C1, C2, E1 C3, C4 and E2 is a really useful terminology for the Central Iberian Zone. We have slightly changed the description of C4 to clarify this point:

**DJ: "The N-S shortening (in present day coordinates) of C3 deformation continued through the Early Permian under brittle conditions (so-called C4 event) (e.g. Dias da Silva et al., in press) and overlapped with the formation of E2 extensional faults (Fig. 2A; Dias and Ribeiro 1991; Dias et al. 2003; Rubio Pascual et al., 2013; Arango et al., 2013; Fernández-Lozano et al. 2019; Dias da Silva et al., in press). C4 N-S compression produced a series of NNE-SSW and NNW-SSE brittle faults (Gil Toja et al. 1985; Dias and Ribeiro 1991; Dias et al. 2003; Fernández-Lozano et al., 2019) and associated sub-vertical and sub-horizontal widespread kink-bands (e.g. Aller et al., 2020; Dias da Silva et al., in press) that are today exposed in Northern Iberia. E2 developed core complex-like structures with extensional shear zones that further telescoped M2 metamorphic isograds between the anatectic cores of gneiss domes

and the associated hanging wall units. This event favored sub-horizontal folding, and crenulation cleavage development in the footwall together with kink-band generation in the upper low-grade structural levels."

"In addition, I also disagree with the notion that the early Permian (i.e. posterior to oroclinal buckling) was characterized by N-S compression. Instead, literature generally agrees that Gondwana-Laurussia convergence during this time period was NW-SE to W-E directed. In the southern Variscan orogeny this led to NW-SE transpression and strike-slip deformation, especially observed in Stephanian-Permian intramontane basins. Simultaneously, to the west of the Variscan orogeny, frontal collision of Gondwana and Laurussia led to the formation of the Alleghenides, Mauritanides and the Ouachita-Marathon-Sonora orogenic belts. The authors do not take into account the related literature, which are not in accordance with their geodynamic model in Fig. 13E. Hence, I would suggest that the authors either (i) limit themselves to the geodynamics of oroclinal buckling (C3 – 315-295Ma) in their interpretation, or (ii) strongly extend their literature study on the kinematics of post-Variscan deformation (295-280Ma)."**

Dominique Jacques is right about the west- and south-wards evolution of the Variscan Alleghanian orogen. In our paper, however we are reviewing exclusively the structural and kinematic evolution of Iberia, where all studied C4 structures (developed in Early to Mid Permian) indicate N-S shortening in present coordinates (which would be WSW-WNW in paleogeographic coordinates). We have added extra references supporting the C4 N-S compression in Iberia. Our kinematic model is restricted to Iberia and it is out of the scope of the paper to fit it within a global or large scale plate reconstruction. We want to remark that strain patterns in Iberia do not necessarily indicate or contradict any particular far field stress. Instead, these strain patterns should be taken into account for regional paleogeographic and global tectonic models, and not the opposite.

**DJ: "What about the stratigraphic successions of the Southern Central Iberian Zone and the West Asturian-Leonese zones, which are connected in the Shaw et al. (2012) model (Fig. 2C)? Can these indeed be connected from this viewpoint, because literature appears to suggest that their lithological succession is different. As your Fig. 3A demonstrates, the lithological succession in the WALZ is much more proximal to the Gondwana shelf than in the CIZ, which consists of more shallow marine lithologies.In a similar fashion, is the metamophic grade of regional metamorphism (mainly active during D1-D2) in both zones comparative?"**

This is a good question that is still unsolved. The stratigraphy of the southermost part of the Central Iberian Zone has been studied in lesser detail, probably because of its poorer exposure. Best stratigraphical sections coming from Almadén area are quite similar to the Tamames area. However, the area covered is small and the correlations are not yet certain. This is another exciting problem yet to be solved. Nevertheless, there is consensus that the stratigraphical architecture is controlled by the presence of structural highs and troughs that caused differences in the shallowness of the deposits and, if so, it cannot be correlated directly with proximity to the Gondwanan shelf. From this point of view, the deeper facies rocks in the WALZ only indicates the presence of a marked through, despite its location within the Gondwanan margin.

―――――――――――――――――――

---

## Author Response (AR2)

Editors of Solid Earth

Dear Editors,

Please find enclosed the corrected version of the paper entitled: "The enigmatic curvature of Central Iberia and its puzzling kinematics." , co-authored by Daniel Pastor-Galán, Gabriel Gutiérrez-Alonso and Arlo B. Weil to be considered in Solid Earth.

We have followed the topical editor comments. As a note, Leite Mendes et al., is accepted (under a minor iteration of corrections, right now). It does not have the final DOI yet, we have provided in the mean-time the EarthArXiv DOI. We hope the final DOI of the paper will be available soon, during the production of this manuscript.

Sincerely,

Daniel Pastor Galán on behalf of all co-authors